



# SymPKF (v1.0): a symbolic and computational toolbox for the design of parametric Kalman filter dynamics

Olivier Pannekoucke[1,2,3] and Philippe Arbogast[4]

[1]INPT-ENM, Toulouse, France.
[2]CNRM, Université de Toulouse, Météo-France, CNRS, Toulouse, France.
[3]CERFACS, Toulouse, France.
[4]Météo-France, Toulouse, France.

**Correspondence:** Olivier Pannekoucke (olivier.pannekoucke@meteo.fr)

**Abstract.** Recent researches in data assimilation lead to the introduction of the parametric Kalman filter (PKF): an implementation of the Kalman filter, where the covariance matrices are approximated by a parameterized covariance model. In the PKF, the dynamics of the covariance during the forecast step relies on the prediction of the covariance parameters. Hence, the design of the parameter dynamics is crucial while it can be tedious to do this by hand. This contribution introduces a Python package, SymPKF, able to compute PKF dynamics for univariate statistics and when the covariance model is parameterized from the variance and the local anisotropy of the correlations. The ability of SymPKF to produce the PKF dynamics is shown on a non-linear diffusive advection (Burgers equation) over a 1D domain and the linear advection over a 2D domain. The computation of the PKF dynamics is performed at a symbolic level, but an automatic code generator is also introduced to perform numerical simulations. A final multivariate example illustrates the potential of SymPKF to go beyond the univariate case.

## 1 Introduction

The Kalman filter (KF) (Kalman, 1960) is one of the backbones of data assimilation. This filter represents the dynamics of a Gaussian distribution all along the analysis and forecast cycles, and takes the form of two equations representing the evolution of the mean and of the covariance of the Gaussian distribution.

While the equations of the KF are simple linear algebra, the large dimension of linear space encountered in the realm of data assimilation make the KF impossible to handle, and this is particularly true for the forecast step. This limitation has motivated some approximation of covariance matrix to make the KF possible. For instance, in ensemble method (Evensen, 2009), the covariance matrix is approximated by a sample estimation, where the time evolution of the covariance matrix is then deduced from the forecast of each individual sample. In the parametric Kalman filter (PKF) (Pannekoucke et al., 2016, 2018b, a), the covariance matrix is approximated by a parametric covariance model, where the time evolution of the matrix is deduced from the time integration of the parameters' evolution equations.





One of the major limitation for the PKF is the design of the parameter evolution equations Although not difficult from a mathematical point of view, this step requires the calculation of many terms that are difficult to calculate by hand and may involve errors in the calculation. To facilitate the derivation of the parametric dynamics and certify the correctness of the resulting system a symbolic derivation of the dynamics would be welcome.

The goal of the package SymPKF 1.0[1] is to facilitate the computation of the PKF dynamics for a particular class of covariance model, the VLATcov model, which is parameterized by the variance and the anisotropy. The symbolic computation of the PKF dynamics relies on a computer algebra system (CAS) able to handle abstract mathematical expressions. A preliminary version has been implemented with Maxima[2] (Pannekoucke, 2021a). However, in order to create an integrated framework which would include the design of the parametric system, as well as its numerical evaluation, the symbolic Python package SymPy (Meurer et al., 2017) has been preferred for the present implementation. In particular, SymPKF comes with an automatic code generator so to provide an end-to-end exploration of the PKF approach, from the computation of the PKF dynamics to its numerical integration.

The paper is organized as follows. The next section provides the background on data assimilation and introduces the PKF. The Section 3 focuses on the PKF for univariate VLATcov models, in the perspective of its symbolic computation by a CAS. Then, the package SymPKF is introduced in Section 4 from its use on the non-linear diffusive advection (the Burgers' equation) over a 1D domain. A numerical example illustrates the use of the automatic code generator provided in SymPKF. Then, the example of the linear advection over a 2D domain shows the ability of SymPKF to handle 2D and 3D domains. The section ends with a simple illustration of a multivariate situation, which also shows that SymPKF applies on a system of prognostic equations. The conclusion is given in Section 5.

## 2 Description of the PKF

### 2.1 Context of the numerical prediction

Dynamics encountered in geosciences is given as a system of partial differential equations

$$\partial_t \mathcal{X} = \mathcal{M}(t, \partial \mathcal{X}), \tag{1}$$

where $\mathcal{X}(t, \mathbf{x})$ is the state of the system and denotes either a scalar field or multivariate fields in a coordinate system $\mathbf{x} = (x^i)_{i \in [1,d]}$ where $d$ is the dimension the geographical space ; $\partial \mathcal{X}$ are the partial derivatives with respect to the coordinate system at any orders, with the convention that order zero denotes the field $\mathcal{X}$ itself ; and $\mathcal{M}$ denotes the trend of the dynamics. A spatial discretization (*e.g.* by using finite differences, finite elements, finite volumes, spectral decomposition, *etc*) transforms Eq. (1) into

$$\partial_t \mathcal{X} = \mathcal{M}(t, \mathcal{X}), \tag{2}$$

---

[1]https://github.com/opannekoucke/sympkf
[2]http://maxima.sourceforge.net/



where this time, $\mathcal{X}(t)$ is a vector, and $\mathcal{M}$ denotes this time the discretization of the trend in Eq. (1). Thereafter, $\mathcal{X}$ can be seen either as a collection of continuous fields with dynamics given by Eq. (1) or a discrete vector of dynamics Eq. (2).

Because of the sparsity and the error of the observations, the forecast $\mathcal{X}^f$ is only an estimation of the true state $\mathcal{X}^t$, which is known to within a forecast-error defined by $e^f = \mathcal{X}^f - \mathcal{X}^t$. This error is often modelled as an unbiased random variable, $\mathbb{E}\left[e^f\right] = 0$. In the discrete formulation of the dynamics Eq. (2), the forecast-error covariance matrix is given by $\mathbf{P}^f = \mathbb{E}\left[e^f(e^f)^{\mathrm{T}}\right]$ where the superscript $^{\mathrm{T}}$ denotes the transpose operator. Since this contribution is focused on the forecast step, thereafter the upper script $^f$ is removed for the sake of simplicity.

We now detail how the error-covariance matrix evolves during the forecast by considering the formalism of the second-order nonlinear Kalman filter.

## 2.2 Second-order nonlinear Kalman filter

A second-order nonlinear Kalman filter (KF2) is a filter that extends the Kalman filter (KF) to the nonlinear situations where the error-covariance matrix evolves tangent-linearly along the trajectory of the mean state and where the dynamics of this mean is governed by the fluctuation-mean interacting dynamics (Jazwinski, 1970; Cohn, 1993). Hence, we first state the dynamics of the mean under the fluctuation-mean interaction, then the dynamics of the error covariance. Note that the choice of the following presentation is motivated by the perspective of using a computer algebra system to perform the computation.

### 2.2.1 Computation of the fluctuation-mean interaction dynamics

Because of the uncertainty on the initial condition, the state $\mathcal{X}$ is modelized as a Markov process $\mathcal{X}(t, \mathbf{x}, \omega)$ where $\omega$ stands for the stochasticity, while $\mathcal{X}$ evolves by Eq. (1). Hence, $\omega$ lies within a certain probability space $(\Omega, \mathcal{F}, P)$ where $\mathcal{F}$ is a $\sigma-$algebra on $\Omega$ (a family of subsets of $\Omega$, which contains $\Omega$ and which is stable for the complement and the countable union) and $P$ is a probability measure see *e.g.* (Øksendal, 2003, chap.2) ; and $\mathcal{X}(t, \mathbf{x}, \cdot) : (\Omega, \mathcal{F}) \to (\mathbb{R}^n, \mathcal{B}_{\mathbb{R}^n})$ is a $\mathcal{F}-$measurable function where $\mathcal{B}_{\mathbb{R}^n}$ denotes the Borel $\sigma-$algebra on $\mathbb{R}^n$, where the integer $n$ is either the dimension of the multivariate field $\mathcal{X}(t, x)$ or the dimension of its discretized version $\mathcal{X}(t)$. The connection between the Markov process and the parameter dynamics is obtained using the Reynolds averaging technique.

So to perform the Reynolds averaging of Eq. (1), the first step is to replace the random field by its Reynolds decomposition $\mathcal{X}(t, \mathbf{x}, \omega) = \mathbb{E}\left[\mathcal{X}\right](t, \mathbf{x}) + \eta e(t, \mathbf{x}, \omega)$. In this modelling of the random state, $\mathbb{E}\left[\mathcal{X}\right]$ is the ensemble average or the mean state; $e$ is an error or a fluctuation to the mean, and it is an unbiased random field, $\mathbb{E}\left[e\right] = 0$. Then, Eq. (1) reads as

$$\partial_t \mathbb{E}\left[\mathcal{X}\right] + \eta \partial_t e = \mathcal{M}(t, \partial \mathbb{E}\left[\mathcal{X}\right] + \eta \partial e), \tag{3}$$

where $\eta$ is a control of magnitude introduced to facilitate Taylor's expansion when using a computer algebra system. At the second order, the Taylor's expansion in $\eta$ of Eq. (3) reads

$$\partial_t \mathbb{E}\left[\mathcal{X}\right] + \eta \partial_t e = \mathcal{M}(t, \partial \mathbb{E}\left[\mathcal{X}\right]) + \eta \mathcal{M}'(t, \partial \mathbb{E}\left[\mathcal{X}\right])(\partial e) + \eta^2 \mathcal{M}''(t, \partial \mathbb{E}\left[\mathcal{X}\right])(\partial e \otimes \partial e), \tag{4a}$$





where $\mathcal{M}'$ and $\mathcal{M}''$ are two linear operators, the former (the later) refers to the tangent-linear model (the hessian), both computed with respect to the mean state $\mathbb{E}[\mathcal{X}]$. The first order expansion is deduced from Eq. (4a) by setting $\eta^2 = 0$, which then reads as

$$\partial_t \mathbb{E}[\mathcal{X}] + \eta \partial_t e = \mathcal{M}(t, \partial \mathbb{E}[\mathcal{X}]) + \eta \mathcal{M}'(t, \partial \mathbb{E}[\mathcal{X}])(\partial e). \quad (4b)$$

By setting $\eta$ to one, the dynamics of the ensemble average is calculated at the second order from the expectation of Eq. (4a) that reads as

$$\partial_t \mathbb{E}[\mathcal{X}] = \mathcal{M}(t, \partial \mathbb{E}[\mathcal{X}]) + \mathcal{M}''(t, \partial \mathbb{E}[\mathcal{X}])(\mathbb{E}[\partial e \otimes \partial e]), \quad (5)$$

where $\partial e \otimes \partial e$ denotes the tensor product of the partial derivatives with respect to the spatial coordinates, *i.e.* terms as $\partial^k e \partial^m e$ for any positive integers $(k, m)$. Here, we have used that the partial derivative commutes with the expectation, $\mathbb{E}[\partial e] = \partial \mathbb{E}[e]$, and that $\mathbb{E}[e] = 0$. Because the expectation is a projector, $\mathbb{E}[\mathbb{E}[\cdot]] = \mathbb{E}[\cdot]$, expectation of $\mathcal{M}(t, \partial \mathbb{E}[\mathcal{X}])$ is itself. The second term of the right hand side makes appear the retro action of the error onto the ensemble averaged dynamics. Hence, Eq. (5) gives the dynamics of the error-mean interaction (or fluctuation-mean interaction).

Note that, the tangent-linear dynamics along the ensemble averaged dynamics Eq. (5) is obtained as the difference between the first order Taylor's expansion Eq. (4b) by its expectation, and reads as

$$\partial_t e = \mathcal{M}'(t, \partial \mathbb{E}[\mathcal{X}])(\partial e). \quad (6)$$

Now it is possible to detail the dynamics of the error covariance from the dynamics of the error which tangent-linearly evolves along the mean state $\mathbb{E}[\mathcal{X}]$.

### 2.2.2 Computation of the error-covariance dynamics

In the discretized form, the dynamics of the error Eq. (6) reads as

$$\frac{de}{dt} = \mathbf{M}e, \quad (7)$$

where $\mathbf{M}$ stands for the tangent-linear model $\mathcal{M}'(t, \partial \mathbb{E}[\mathcal{X}])$ evaluated at the mean state $\mathbb{E}[\mathcal{X}]$. So the dynamics of the error-covariance matrix, $\mathbf{P} = \mathbb{E}[ee^{\mathrm{T}}]$, is given by

$$\frac{d\mathbf{P}}{dt} = \mathbf{MP} + \mathbf{PM}^{\mathrm{T}} \quad (8a)$$

($\mathbf{M}^{\mathrm{T}}$ is the adjoint of $\mathbf{M}$), or its integrated version

$$\mathbf{P}(t) = \mathbf{M}_{t\leftarrow 0}\mathbf{P}_0 (\mathbf{M}_{t\leftarrow 0})^{\mathrm{T}} \quad (8b)$$

where $\mathbf{M}_{t\leftarrow 0}$ is the propagator associated to the time integration of Eq. (7), initiated from the covariance $\mathbf{P}_0$.





### 2.2.3 Setting of the KF2

Gathering the dynamics of the ensemble mean given by the fluctuation-mean interaction Eq. (5) and the covariance dynamics Eq. (8) leads to the second-order closure approximation of the extended KF, that is the forecast step equations of the KF2.

Similarly to the KF, the principal limitation of the KF2 is the numerical cost associated with the covariance dynamics Eq. (8): living in a discrete world, the numerical cost of Eq. (8) dramatically increases with the size of the problem. As an example, for the dynamics of simple scalar field discretized with $n$ grid points, the dimension of its vector representation is $n$, while the size

of the error-covariance matrix scales as $n^2$ ; leading to a numerical cost of Eq. (8) between $\mathcal{O}(n^2)$ and $\mathcal{O}(n^3)$.

We now introduce the parametric approximation of covariance matrices which aims to reduce the cost of the covariance dynamics Eq. (8).

### 2.3 Formulation of the PKF prediction

The parametric formulation of covariance evolution stands as follows. If $\mathbf{P}(\mathcal{P})$ denotes a covariance model featured by a set of

120 parameters $\mathcal{P} = (p_i)_{i \in I}$, then there exists a set $\mathcal{P}_t^f$ featuring the forecast error covariance matrix so that $\mathbf{P}(\mathcal{P}_t^f) \approx \mathbf{P}_t^f$.

Hence, starting from the initial condition $\mathcal{P}^f = \mathcal{P}_0^f$, if the dynamics of the parameters $\mathcal{P}_t^f$ is known, then it is possible to approximately determine $\mathbf{P}_t^f \approx \mathbf{P}(\mathcal{P}_t^f)$ without solving Eq. (8) explicitly. This approach constitutes the so-called parametric Kalman filter (PKF) approximation, introduced by Pannekoucke et al. (2016, 2018a) (P16, P18).

We now focus on the PKF applied to a particular family of covariance models.

### 3 PKF for VLATcov models

This part introduces a particular family of covariance models, parameterized by the fields of variances and of local anisotropy tensor: the VLATcov models (Pannekoucke, 2021b). What makes this covariance model interesting is that its parameters are related to the error field and thus, it is possible to determine the dynamics of the parameters. So to introduce VLATcov models, we first present the diagnosis of the variance and of the local anisotropy tensor, then we present two examples of VLATcov

models and we end the section by the description of the dynamics of the parameters.

### 3.1 Definition of the fields of variance and of local anisotropy tensor

From now, we will focus on the forecast-error statistics, so the upper script $^f$ is removed for the sake of simplicity. Moreover, for a function $f$, when there is no confusion, the value of $f$ at a point $\mathbf{x}$ is written either as $f(\mathbf{x})$ or as $f_\mathbf{x}$.

The forecast error being unbiased, $\mathbb{E}[e] = 0$, its variance at a point $\mathbf{x}$ is defined as

$$V(\mathbf{x}) = \mathbb{E}\left[e(\mathbf{x})^2\right]. \tag{9}$$





When the error is a random differentiable field, the anisotropy of the two-points correlation function $\rho(\mathbf{x}, \mathbf{y}) = \frac{1}{\sqrt{V_\mathbf{x} V_\mathbf{y}}}\mathbb{E}\left[e(\mathbf{x})e(\mathbf{y})\right]$ is featured, from the second order expansion

$$\rho(\mathbf{x}, \mathbf{x} + \delta\mathbf{x}) \approx 1 - \frac{1}{2}||\delta\mathbf{x}||^2_{\mathbf{g}_\mathbf{x}}, \tag{10}$$

by the local metric tensor $\mathbf{g}(\mathbf{x})$, and defined as

$$\mathbf{g}(\mathbf{x}) = -\nabla\nabla^{\mathrm{T}}\rho_\mathbf{x}, \tag{11}$$

where $\rho_\mathbf{x}(\mathbf{y}) = \rho(\mathbf{x}, \mathbf{y})$ *e.g.*

$$g_{ij}(\mathbf{x}) = -\left(\partial^2_{y^i y^j}\rho_\mathbf{x}(\mathbf{y})\right)_{\mathbf{y}=\mathbf{x}}.$$

The metric tensor is a symmetric positive definite matrix, and it is a $2 \times 2$ ($3 \times 3$) matrix in a 2D (3D) domain.

Note that it is useful to introduce the local aspect tensor (Purser et al., 2003), whose the geometry goes as the correlation, and defined as the inverse of the metric tensor

$$\mathbf{s}(\mathbf{x}) = \mathbf{g}(\mathbf{x})^{-1}, \tag{12}$$

where the superscript $^{-1}$ denotes the matrix inverse.

What makes the metric tensor attractive, either at a theoretical or at a practical level, is that it is closely related to the normalized error $\varepsilon = \frac{e}{\sqrt{V}}$ by

$$\mathbf{g}_{ij}(\mathbf{x}) = \mathbb{E}\left[(\partial_{x^i}\varepsilon)(\partial_{x^j}\varepsilon)\right] \tag{13}$$

(see *e.g.* (Pannekoucke, 2021b) for details).

Hence, a VLATcov model is a covariance model characterized by the variance field, and by the anisotropy field, the latter being defined either by the metric-tensor field $\mathbf{g}$ or by the aspect-tensor field $\mathbf{s}$. To put some flesh on the bones, two examples of VLATcov models are now presented.

## 3.2 Examples of VLATcov models

We first consider the covariance model based on the heterogeneous diffusion operator of Weaver and Courtier (2001), which
is used in variational data assimilation to model heterogeneous correlation functions *e.g.* for the ocean or for air quality. This model has the property that, under the local homogenous assumption (that is when the spatial derivatives are negligible) the local aspect tensors of the correlation functions are twice the local diffusion tensors. (Pannekoucke and Massart, 2008; Mirouze and Weaver, 2010). Hence, by defining the local diffusion tensors as half the local aspect tensors, the covariance model based on the heterogeneous diffusion equation is a VLATcov model.

Another example of heterogeneous covariance model is the heterogeneous Gaussian covariance model

$$\mathbf{P}^{\mathrm{he \cdot g}}(V, \nu)(\mathbf{x}, \mathbf{y}) = \sqrt{V(\mathbf{x})V(\mathbf{y})}\frac{|\nu_\mathbf{x}|^{1/4}|\nu_\mathbf{y}|^{1/4}}{\left|\frac{1}{2}(\nu_\mathbf{x} + \nu_\mathbf{y})\right|^{1/2}}\exp\left(-||\mathbf{x} - \mathbf{y}||^2_{(\nu_\mathbf{x} + \nu_\mathbf{y})^{-1}}\right), \tag{14}$$





where $\nu$ is a field of symmetric positive definite matrices, and $|\nu|$ denotes the matrix determinant. $\mathbf{P}^{\mathrm{he.g}}(V,\nu)$ is a particular case of the class of covariance models deduced from Theorem 1 of Paciorek and Schervish (2004). Again, this covariance

model has the property that, under local homogenous assumptions, the local aspect tensor is approximately given by $\nu$, *i.e.* for any point $\mathbf{x}$,

$$\mathbf{s}_x \approx \nu_x. \tag{15}$$

Hence, as for the covariance model based on the diffusion equation, by defining the field $\nu$ as the aspect tensor field, the heterogeneous Gaussian covariance model is a VLATcov model (Pannekoucke, 2021b).

At this stage, all the pieces of the puzzle are put together to build the PKF dynamics. We have covariance models parameterized from the variance and the local anisotropy, which are both related to the error field: knowing the dynamics of the error leads to the dynamics of the VLATcov parameters. This is now detailed.

### 3.3 PKF prediction step for VLATcov models

When the dynamics of the error $e$ is well approximated from the tangent-linear evolution Eq. (6), the connection between

175 the covariance parameters and the error, Eq. (9) and Eq. (13), makes possible to establish the prediction step of the PKF (Pannekoucke et al., 2018a), which reads as the dynamics of the ensemble average (at the second-order closure)

$$\partial_t \mathbb{E}\left[\mathcal{X}\right] = \mathcal{M}(t, \partial \mathbb{E}\left[\mathcal{X}\right]) + \mathcal{M}''(t, \partial \mathbb{E}\left[\mathcal{X}\right])(\mathbb{E}\left[\partial e \otimes \partial e\right]), \tag{16a}$$

coupled with the dynamics of the variance and the metric

$$\partial_t V(t, \mathbf{x}) = 2\mathbb{E}\left[e\partial_t e\right], \tag{16b}$$

$$\partial_t \mathbf{g}_{ij}(t, \mathbf{x}) = \mathbb{E}\left[\partial_t \left((\partial_{x^i}\varepsilon)(\partial_{x^j}\varepsilon)\right)\right], \tag{16c}$$

where it remains to replace the dynamics of the error (and its normalized version $\varepsilon = e/\sqrt{V}$) from Eq. (6), and where property that the expectation operator and the temporal derivative commutes, $\partial_t \mathbb{E}\left[\cdot\right] = \mathbb{E}\left[\partial_t \cdot\right]$, has been used to obtain Eq. (16b) and Eq. (16c).

The set of equations (16) is at the heart of the numerical sobriety of the parametric approach. In contrast to the matrix dynamics of the KF, the PKF approach is designed for the continuous world, leading to PDEs for the parameter dynamics in place of ODEs Eq. (8) for the full matrix dynamics. For the above mentioned scalar fields, introduced is the computation of the algorithmic complexity in section 2.1, the cost of Eq. (16) is $\mathcal{O}(n)$. Moreover, the dynamics of the parameters sheds light on the nature of the processes governing the dynamics of covariances ; and it does not require any adjoint of the dynamics

(Pannekoucke et al., 2016, 2018a).

Note that Eq. (16) can be formulated in terms of aspect tensors, thanks to the definition Eq. (12): since, $\mathbf{sg} = \mathbf{I}$, its time derivative $(\partial_t \mathbf{s})\mathbf{g} + \mathbf{s}(\partial_t \mathbf{g}) = 0$ leads to the dynamics $\partial_t \mathbf{s} = -\mathbf{g}^{-1}(\partial_t \mathbf{g})\mathbf{s}$, and then

$$\partial_t \mathbf{s} = -\mathbf{s}(\partial_t \mathbf{g})\mathbf{s}, \tag{17}$$





where it remains to replace occurrences of $\mathbf{g}$ by $\mathbf{s}^{-1}$ in the resulting dynamics of the mean, the variance and the aspect tensor.

Hence, the PKF forecast step for VLATcov model is given by either the system Eq. (16) (in metric), or by its aspect tensor formulation thanks to Eq. (17). Whatever the formulation considered, it is possible to carry out the calculations using a formal calculation language. However, even for simple physical processes, the number of terms in formal expressions can become very large, *e.g.* it is common to have to manipulate expressions with more than a hundred terms. Thus, any strategy that simplifies the assessment of PKF systems in advance can quickly become a significant advantage.

In the following section, we present the splitting method that allows the PKF dynamics to be expressed by bringing together the dynamics of each of the physical processes, calculated individually.

### 3.4   The splitting strategy

When there are several processes in the dynamics Eq. (1), the calculation of the parametric dynamics can be tedious even when using a computer algebra system. To better use digital resources, a splitting strategy can be introduced (Pannekoucke et al.,

2016, 2018a).

While the theoretical background is provided by the Lie-Trotter formula for Lie derivatives, the well-known idea of time-splitting is easily catched from a first order Taylor expansion of an Euler numerical scheme:

The computation of a dynamics

$$\partial_t \mathcal{X} = f_1(\mathcal{X}) + f_2(\mathcal{X}), \tag{18}$$

over a single time step $\delta t$, can be done in two times following the numerical scheme

$$\begin{cases} \mathcal{X}^\star = \mathcal{X}(t) + \delta t f_1(\mathcal{X}(t)), \\ \mathcal{X}(t+\delta t) = \mathcal{X}^\star + \delta t f_2(\mathcal{X}^\star). \end{cases} \tag{19}$$

where at order $\delta t$, this scheme is equivalent to $\mathcal{X}(t+\delta t) = \mathcal{X}(t) + \delta t (f_1(\mathcal{X}(t)) + f_2(\mathcal{X}(t)))$, that is the Euler step of Eq. (18). Because $f_1$ and $f_2$ can be viewed as vector fields, the fractional scheme, joining the starting point (at $t$) to the end point (at $t + \delta t$), remains to going through the parallelogram, formed by the sum of the two vectors, along its sides. Since there are two

paths joining the extreme points, starting the computation by $f_2$ is equivalent to starting by $f_1$ (at order $\delta t$), this corresponds to the commutativity of the diagram formed by the parallelogram.

Appendix A shows that a dynamics given by Eq. (18) implies a dynamics of the error, the variance, the metric and the aspect, written as a sum of trends. Hence, it is possible to apply a splitting for all these dynamics.

As a consequence for the calculation of the parametric dynamics: calculating the parametric dynamics of Eq. (18) is equiv-

alent to calculating separately the parametric dynamics of $\partial_t \mathcal{X} = f_1(\mathcal{X})$ and $\partial_t \mathcal{X} = f_2(\mathcal{X})$, then bringing together the two parametric dynamics into a single one by summing the trends for the mean, the variance, the metric or the aspect dynamics. This splitting applies when there are more than two processes and appears as a general method to reduce the complexity of the calculation.





### 3.5 Discussion/intermediate conclusion

However, although the calculation of the system Eq. (16) is straightforward, as it is similar to the calculation of Reynolds equations (Pannekoucke et al., 2018a), it is tedious because of the many terms involved, and there is a risk of introducing errors during the calculation by hand.

Then, once the dynamics of the parameters is established, it remains to design a numerical code to test whether the uncertainty is effectively well represented by the PKF dynamics. Again, the design of a numerical code is not necessarily difficult
but with the numerous terms, the risk of introducing an error is important.

To facilitate the design of the PKF dynamics as well as the numerical evaluation, the package SymPKF has been introduced to perform the VLATcov parameter dynamics and to generate a numerical code used for the investigations (Pannekoucke, 2021c). The next section introduces and details this tool.

## 4 Symbolic computation of the PKF for VLATcov

In order to introduce the symbolic computation of the PKF for VLATcov model, we consider an example: the diffusive non-linear advection, the Burgers equation, which reads as

$$\partial_t u + u \partial_x u = \kappa \partial_x^2 u, \tag{20}$$

where $u$ stands for the velocity field and corresponds to a function of the time $t$ and the space of coordinate $x$, and where $\kappa$ is a diffusion coefficient (constant here). This example illustrates the workflow leading to the PKF dynamics. It consists in defining
the system of equations in SymPy, then to compute the dynamics Eq. (16), we now detail these two steps.

### 4.1 Definition of the dynamics

The definition of the dynamics relies on the formalism of SymPy as shown in Fig. 1. The coordinate system is first defined as instances of the class `Symbols`. Note that the time is defined as `sympkf.t` while the spatial coordinate is let to the choice of the user, here $x$. Then, the function $u$ is defined as an instance of the class `Function`, as a function of $(t, x)$.
In this example, the dynamics consists in a single equation defined as an instance of the class `Eq`, but in the general situation where the dynamics is given as a system of equations, the dynamics has to be represented as a Python list of equations.

A preprocessing of the dynamics is then performed to determine several important quantities to handle the dynamics: the prognostic fields (functions for which a time derivative is present), the diagnostic fields (functions for which there is no time derivative in the dynamics), the constant functions (functions that only depend on the spatial coordinates), and the constants
(pure scalar terms, that are not function of any coordinate). This preprocessing is performed when transforming the dynamics as an instance of the class `PDESystem`, and whose default string output delivers a summary of the dynamics: for the Burgers' equation, there is only one prognostic function, $u(t, x)$, and one constant, $\kappa$.

The prognostic quantities being known, it is then possible to perform the computation of the PKF dynamics, as discussed now.





```python
# Import of libraries
from sympy import symbols, Function, Derivative, Eq
from sympkf import PDESystem, SymbolicPKF, t

# Set the spatial coordinate system
x = symbols('x')
# Set the constants
kappa = symbols('kappa')
# Define the spatio-temporal scalar field
u = Function('u')(t,x)
```

```python
# Definition of the Burgers dynamics
burgers_equation = Eq(Derivative(u,t),
    -u*Derivative(u,x)+kappa*Derivative(u,x,2))
burgers_equation
```

$$\frac{\partial}{\partial t}u(t,x) = \kappa\frac{\partial^2}{\partial x^2}u(t,x) - u(t,x)\frac{\partial}{\partial x}u(t,x)$$

```python
# Processing of the PDE system
burgers = PDESystem( burgers_equation )
burgers
```

```
PDE System :
        prognostic functions : u(t, x)
        constant functions   :
        exogeneous functions :
        constants            : kappa
```

**Figure 1.** Sample of code and Jupyter notebook outputs for the definition of the Burgers dynamics using SymPKF.

## 4.2 Computation of the VLATcov PKF dynamics

Thanks to the preprocessing, we are able to determine what are the VLATcov parameters needed to compute the PKF dynamics, that is the variance and the anisotropy tensor associated to the prognostic fields. For the Burgers' equation, the VLATcov parameters are the variance $V_u$ and the metric tensor $\mathbf{g}_u = (g_{u,xx})$ or its associated aspect tensor $\mathbf{s}_u = (s_{u,xx})$. Note that, in SymPKF, the VLATcov parameters are labelled by their corresponding prognostic fields so to facilitate their identification. This labelling is achieved when the dynamics is transformed as an instance of the class `SymbolicPKF`. This class is at the core of the computation of the PKF dynamics from Eq. (16).

As discussed in Section 2.2.1, the PKF dynamics relies on the second-order flucutation-mean interaction dynamics where each prognostic function is replaced by a stochastic counter-part. Hence, the constructor of `SymbolicPKF` converts each prognostic functions as a function of an additional coordinate, $\omega \in \Omega$. For the Burgers' equation, $u(t,x)$ becomes $u(t,x,\omega)$.

Since the computation of the second-order fluctuation-mean interaction dynamics relies on the expectation operator, an implementation of this expectation operator has been introduced in SymPKF: it is defined as the class `Expectation` build by inheritance from the class `sympy.Function` so to leverage on the computational facilities of SymPy. The implementation of the class `Expectation` is based on the linearity of the mathematical expectation operator with respect to deterministic quantities, and its commutativity with partial derivatives and integrals with respect to coordinates different from $\omega$ *e.g.* for



```
pkf_burgers = SymbolicPKF(burgers)
```

```
for equation in pkf_burgers.in_metric: display(equation)
```

$$\frac{\partial}{\partial t}u(t,x) = \kappa \frac{\partial^2}{\partial x^2}u(t,x) - u(t,x)\frac{\partial}{\partial x}u(t,x) - \frac{\frac{\partial}{\partial x}V_u(t,x)}{2}$$

$$\frac{\partial}{\partial t}V_u(t,x) = -2\kappa V_u(t,x) g_{u,xx}(t,x) + \kappa \frac{\partial^2}{\partial x^2}V_u(t,x) - \frac{\kappa\left(\frac{\partial}{\partial x}V_u(t,x)\right)^2}{2V_u(t,x)} - u(t,x)\frac{\partial}{\partial x}V_u(t,x) -$$
$$2V_u(t,x)\frac{\partial}{\partial x}u(t,x)$$

$$\frac{\partial}{\partial t}g_{u,xx}(t,x) = 2\kappa g_{u,xx}{}^2(t,x) - 2\kappa\mathbb{E}\left(\varepsilon_u(t,x,\omega)\frac{\partial^4}{\partial x^4}\varepsilon_u(t,x,\omega)\right) - 3\kappa\frac{\partial^2}{\partial x^2}g_{u,xx}(t,x) +$$
$$\frac{2\kappa g_{u,xx}(t,x)\frac{\partial^2}{\partial x^2}V_u(t,x)}{V_u(t,x)} + \frac{\kappa\frac{\partial}{\partial x}V_u(t,x)\frac{\partial}{\partial x}g_{u,xx}(t,x)}{V_u(t,x)} - \frac{2\kappa g_{u,xx}(t,x)\left(\frac{\partial}{\partial x}V_u(t,x)\right)^2}{V_u{}^2(t,x)} -$$
$$u(t,x)\frac{\partial}{\partial x}g_{u,xx}(t,x) - 2g_{u,xx}(t,x)\frac{\partial}{\partial x}u(t,x)$$

```
for equation in pkf_burgers.in_aspect: display(equation)
```

$$\frac{\partial}{\partial t}u(t,x) = \kappa \frac{\partial^2}{\partial x^2}u(t,x) - u(t,x)\frac{\partial}{\partial x}u(t,x) - \frac{\frac{\partial}{\partial x}V_u(t,x)}{2}$$

$$\frac{\partial}{\partial t}V_u(t,x) = -\frac{2\kappa V_u(t,x)}{s_{u,xx}(t,x)} + \kappa\frac{\partial^2}{\partial x^2}V_u(t,x) - \frac{\kappa\left(\frac{\partial}{\partial x}V_u(t,x)\right)^2}{2V_u(t,x)} - u(t,x)\frac{\partial}{\partial x}V_u(t,x) - 2V_u(t,x)\frac{\partial}{\partial x}u(t,x)$$

$$\frac{\partial}{\partial t}s_{u,xx}(t,x) = 2\kappa s_{u,xx}{}^2(t,x)\mathbb{E}\left(\varepsilon_u(t,x,\omega)\frac{\partial^4}{\partial x^4}\varepsilon_u(t,x,\omega)\right) - 3\kappa\frac{\partial^2}{\partial x^2}s_{u,xx}(t,x) - 2\kappa + \frac{6\kappa\left(\frac{\partial}{\partial x}s_{u,xx}(t,x)\right)^2}{s_{u,xx}(t,x)} -$$
$$\frac{2\kappa s_{u,xx}(t,x)\frac{\partial^2}{\partial x^2}V_u(t,x)}{V_u(t,x)} + \frac{\kappa\frac{\partial}{\partial x}V_u(t,x)\frac{\partial}{\partial x}s_{u,xx}(t,x)}{V_u(t,x)} + \frac{2\kappa s_{u,xx}(t,x)\left(\frac{\partial}{\partial x}V_u(t,x)\right)^2}{V_u{}^2(t,x)} - u(t,x)\frac{\partial}{\partial x}s_{u,xx}(t,x) + 2s_{u,xx}(t,x)\frac{\partial}{\partial x}u(t,x)$$

**Figure 2.** Sample of code and Jupyter notebook outputs: systems of partial differential equations given in metric and in aspect forms, produced by SymPKF when applied to the Burgers' equation Eq. (20).

the Burgers' equation $\mathbb{E}[\partial_x u(t,x,\omega)] = \partial_x \mathbb{E}[u(t,x,\omega)]$. Note that $\mathbb{E}[u(t,x,\omega)]$ is a function of $(t,x)$ only: the expectation operator converts a random variable into a deterministic variable.

Then, the symbolic computation of the second-order fluctuation-mean interaction dynamics Eq. (16a) is performed, thanks to SymPy, by following the steps as described in Section 2.2.1. In particular, the computation also leads to the tangent-linear dynamics of the error Eq. (6), from which it is possible to compute the dynamics of the variance Eq. (16b) and of the metric
tensor Eq. (16c) (or its associated aspect-tensor version). Applying these steps, and the appropriate substitutions, is achieved when calling the `in_metric` or `in_aspect` Python's property of an instance of the class `SymbolicPKF`. This is shown for the Burgers' equation in Fig. 2, where the background computation of the PKF dynamics leads to a list of the three coupled equations corresponding to the mean, the variance and the aspect tensor, similar to the system Eq. (22) first obtained by Pannekoucke et al. (2018a).





```python
# PKF for the non-linear advection
nladvection_pkf = SymbolicPKF(
    Eq(Derivative(u,t), -u*Derivative(u,x))
)

for equation in nladvection_pkf.in_aspect: display(equation)
```

$$\frac{\partial}{\partial t}u(t,x) = -u(t,x)\frac{\partial}{\partial x}u(t,x) - \frac{\frac{\partial}{\partial x}\mathrm{V_u}(t,x)}{2}$$

$$\frac{\partial}{\partial t}\mathrm{V_u}(t,x) = -u(t,x)\frac{\partial}{\partial x}\mathrm{V_u}(t,x) - 2\,\mathrm{V_u}(t,x)\frac{\partial}{\partial x}u(t,x)$$

$$\frac{\partial}{\partial t}\mathrm{s_{u,xx}}(t,x) = -u(t,x)\frac{\partial}{\partial x}\mathrm{s_{u,xx}}(t,x) + 2\,\mathrm{s_{u,xx}}(t,x)\frac{\partial}{\partial x}u(t,x)$$

```python
# PKF for the diffusion
diffusion_pkf = SymbolicPKF(
    Eq(Derivative(u,t), kappa*Derivative(u,x,2))
)

for equation in diffusion_pkf.in_aspect: display(equation)
```

$$\frac{\partial}{\partial t}u(t,x) = \kappa\frac{\partial^2}{\partial x^2}u(t,x)$$

$$\frac{\partial}{\partial t}\mathrm{V_u}(t,x) = -\frac{2\kappa\,\mathrm{V_u}(t,x)}{\mathrm{s_{u,xx}}(t,x)} + \kappa\frac{\partial^2}{\partial x^2}\mathrm{V_u}(t,x) - \frac{\kappa\left(\frac{\partial}{\partial x}\mathrm{V_u}(t,x)\right)^2}{2\,\mathrm{V_u}(t,x)}$$

$$\frac{\partial}{\partial t}\mathrm{s_{u,xx}}(t,x) = 2\kappa\,\mathrm{s_{u,xx}}^2(t,x)\mathbb{E}\left(\varepsilon_u(t,x,\omega)\frac{\partial^4}{\partial x^4}\varepsilon_u(t,x,\omega)\right) - 3\kappa\frac{\partial^2}{\partial x^2}\mathrm{s_{u,xx}}(t,x) - 2\kappa +$$

$$\frac{6\kappa\left(\frac{\partial}{\partial x}\mathrm{s_{u,xx}}(t,x)\right)^2}{\mathrm{s_{u,xx}}(t,x)} - \frac{2\kappa\,\mathrm{s_{u,xx}}(t,x)\frac{\partial^2}{\partial x^2}\mathrm{V_u}(t,x)}{\mathrm{V_u}(t,x)} + \frac{\kappa\frac{\partial}{\partial x}\mathrm{V_u}(t,x)\frac{\partial}{\partial x}\mathrm{s_{u,xx}}(t,x)}{\mathrm{V_u}(t,x)} + \frac{2\kappa\,\mathrm{s_{u,xx}}(t,x)\left(\frac{\partial}{\partial x}\mathrm{V_u}(t,x)\right)^2}{\mathrm{V_u}^2(t,x)}$$

**Figure 3.** Illustration of the splitting strategy which can be used to compute the PKF dynamics and applied here for the Burgers' equation: PKF dynamics of the Burgers' equation can be obtained from the PKF dynamics of the advection (first cell) and of the diffusion (second cell).

 

Hence, from SymPKF, for the Burgers' equation, the VLATcov PKF dynamics given in aspect tensor reads as

$$
\begin{cases}
\partial_t u & = \quad \kappa\partial_x^2 u - u\partial_x u - \frac{\partial_x V_u}{2} \\
\partial_t V_u & = \quad -\frac{2\kappa V_u}{s_{u,xx}} + \kappa\partial_x^2 V_u - \frac{\kappa(\partial_x V_u)^2}{2V_u} \\
& \quad\quad -u\partial_x V_u - 2V_u\partial_x u \\
\partial_t s_{u,xx} & = \quad 2\kappa s_{u,xx}^2 \mathbb{E}\left(\varepsilon_u\partial_x^4\varepsilon_u\right) - 3\kappa\partial_x^2 s_{u,xx} \\
& \quad\quad -2\kappa + \frac{6\kappa(\partial_x s_{u,xx})^2}{s_{u,xx}} - \frac{2\kappa s_{u,xx}\partial_x^2 V_u}{V_u} \\
& \quad\quad +\frac{\kappa\partial_x V_u\partial_x s_{u,xx}}{V_u} + \frac{2\kappa s_{u,xx}(\partial_x V_u)^2}{V_u^2} \\
& \quad\quad -u\partial_x s_{u,xx} + 2s_{u,xx}\partial_x u
\end{cases}
\tag{21}
$$

where here $s_{u,xx}$ is the single component of the aspect tensor $\mathbf{s}_u$ in 1D domains. Note that in the output of the PKF equations, as reproduced in Eq. (21), the expectation in the dynamics of the mean is replaced by the prognostic field, that is for the Burgers' equation: $\mathbb{E}[u](t,x)$ is simply denoted by $u(t,x)$.

While the Burgers' equation only contains two physical processes *i.e.* the non-linear advection and the diffusion, the resulting PKF dynamics Eq. (21) makes appear numerous terms, which justifies the use of symbolic computation, as above mentioned. The computation of the PKF dynamics leading to the metric and to the aspect tensor formulation takes about $1s$ of computation (Intel Core i7-7820HQ CPU at 2.90GHz x 8).

In this example, the splitting strategy has not been considered to simplify the computation of the PKF dynamics. However, it can be done by considering the PKF dynamics for the advection $\partial_t u = -u\partial_x u$ and the diffusion $\partial_t u = \kappa\partial_x^2 u$, and computed separately, then merged to find the PKF dynamics of the full Burgers' equation. For instance, the Fig. 3 show the PKF dynamics for the advection (first cell) and for the diffusion (second cell), where the output can be traced back in Eq. (2) *e.g.* by the terms in $\kappa$ for the diffusion.

Thanks to the symbolic computation using the expectation operator, as implemented by the class `Expectation`, it is possible to handle terms as $\mathbb{E}\left[\varepsilon_u\partial_x^4\varepsilon_u\right]$ during the computation of the PKF dynamics. The next section details how these terms are handled during the computation and the closure issue they bring.

### 4.3 Comments on the computation of the VLATcov PKF dynamics and the closure issue

### 4.3.1 Computation of terms $\mathbb{E}\left[\partial^\alpha\varepsilon\partial^\beta\varepsilon\right]$ and their connection to the correlation function

An important point is that terms as $\mathbb{E}[\varepsilon\partial^\alpha\varepsilon]$, *e.g.* $\mathbb{E}\left[\varepsilon_u\partial_x^4\varepsilon_u\right]$ in Eq. (21), are directly connected to the correlation function $\rho(\mathbf{x},\mathbf{y}) = \mathbb{E}[\varepsilon(\mathbf{x})\varepsilon(\mathbf{y})]$ whose Taylor expansion is written as

$$
\rho(\mathbf{x},\mathbf{x}+\delta\mathbf{x}) = \sum_k \frac{1}{k!}\mathbb{E}\left[\varepsilon(\mathbf{x})\partial^k\varepsilon(\mathbf{x})\right]\delta\mathbf{x}^k.
\tag{22}
$$

However, during its computation, the VLATcov PKF dynamics makes appear terms $\mathbb{E}\left[\partial^\alpha\varepsilon\partial^\beta\varepsilon\right]$ with $|\alpha| \leq |\beta|$, where for any $\alpha$, $\partial^\alpha$ denotes the derivative with respect to the multi-index $\alpha = (\alpha_i)_{i\in[1,n]}$, $\alpha_i$ denoting the derivative order with respect to the $i^{th}$ coordinate $x_i$ of the coordinate system ; and where the sum of all derivative order is denoted by $|\alpha| = \sum_i \alpha_i$. The issue it that these terms in $\mathbb{E}\left[\partial^\alpha\varepsilon\partial^\beta\varepsilon\right]$ are not directly connected to the Taylor expansion Eq. (22).



```
for key, value in pkf_burgers.subs_tree.items():
    display({key:value})
```

$$\left\{ \mathbb{E}\left( \left( \frac{\partial}{\partial x}\varepsilon_u(t,x,\omega) \right)^2 \right) : g_{u,xx}(t,x) \right\}$$

$$\left\{ \mathbb{E}\left( \frac{\partial}{\partial x}\varepsilon_u(t,x,\omega)\frac{\partial^2}{\partial x^2}\varepsilon_u(t,x,\omega) \right) : \frac{\frac{\partial}{\partial x}g_{u,xx}(t,x)}{2} \right\}$$

$$\left\{ \mathbb{E}\left( \frac{\partial}{\partial x}\varepsilon_u(t,x,\omega)\frac{\partial^3}{\partial x^3}\varepsilon_u(t,x,\omega) \right) : -\mathbb{E}\left( \varepsilon_u(t,x,\omega)\frac{\partial^4}{\partial x^4}\varepsilon_u(t,x,\omega) \right) - \frac{3\frac{\partial^2}{\partial x^2}g_{u,xx}(t,x)}{2} \right\}$$

$$\left\{ \mathbb{E}\left( \left( \frac{\partial^2}{\partial x^2}\varepsilon_u(t,x,\omega) \right)^2 \right) : \mathbb{E}\left( \varepsilon_u(t,x,\omega)\frac{\partial^4}{\partial x^4}\varepsilon_u(t,x,\omega) \right) + 2\frac{\partial^2}{\partial x^2}g_{u,xx}(t,x) \right\}$$

**Figure 4.** Substitution dictionary computed in SymPKF to replace terms as $\mathbb{E}\left[ \partial^\alpha\varepsilon\partial^\beta\varepsilon \right]$ by terms in $\mathbb{E}\left[ \varepsilon\partial^\gamma\varepsilon \right]$ where $|\gamma| < |\alpha| + |\beta|$.

The interesting property of these terms is that they can be reworded as spatial derivatives of terms in the form $\mathbb{E}[\varepsilon\partial^\gamma\varepsilon]$. More precisely, any term $\mathbb{E}\left[ \partial^\alpha\varepsilon\partial^\beta\varepsilon \right]$ can be written from derivative of terms in $\mathbb{E}[\varepsilon\partial^\gamma\varepsilon]$ where $|\gamma| < |\alpha| + |\beta|$, and the term $\mathbb{E}\left[ \varepsilon\partial^{\alpha+\beta}\varepsilon \right]$ (see Appendix B for the proof). So, to replace any term in $\mathbb{E}\left[ \partial^\alpha\varepsilon\partial^\beta\varepsilon \right]$ by terms in $\mathbb{E}[\varepsilon\partial^\gamma\varepsilon]$ where $|\gamma| < |\alpha| + |\beta|$, a substitution dictionary is computed in SymPKF, and stored as the variable `subs_tree`. The computation of this substitution dictionary is performed thanks to a dynamical programming strategy. Latter, the integer $|\alpha| + |\beta|$ is called the order of the term $\mathbb{E}\left[ \partial^\alpha\varepsilon\partial^\beta\varepsilon \right]$. Fig. 4 shows the substitution dictionary computed for the Burgers' equation. It appears that terms of order lower than 3 can be explicitly written from the metric (or its derivatives) while terms of order larger than 4 cannot: this is known as the closure issue (Pannekoucke et al., 2018a).

The term $\mathbb{E}\left[ \varepsilon\partial_x^4\varepsilon \right]$, which features long-range correlations, cannot be related neither to the variance nor to the metric, and has to be closed. We detail this point in the next section.

### 4.3.2 Analytical and data-driven closure

A naïve closure for the PKF dynamics Eq. (21) would be to replace the unknown term $\mathbb{E}\left[ \varepsilon_u\partial_x^4\varepsilon_u \right]$ by zero. However, in the third equation that corresponds to the aspect tensor dynamics, the coefficient $-3\kappa$ of the diffusion term $\partial_x^2 s_u$ being negative, it follows that the dynamics of $s_u$ numerically explodes at an exponential rate. Of course, because the system represents the uncertainty dynamics of the Burgers' equation Eq. (20) that is well-posed, the parametric dynamics should not explode. Hence, the unknown term $\mathbb{E}\left[ \varepsilon_u\partial_x^4\varepsilon_u \right]$ is crucial: it can balance the negative diffusion so to stabilize the parametric dynamics.

For the Burgers' equation, a closure for $\mathbb{E}\left[ \varepsilon_u\partial_x^4\varepsilon_u \right]$ has been previously proposed (Pannekoucke et al., 2018a), given by

$$\mathbb{E}\left[ \varepsilon_u\partial_x^4\varepsilon_u \right] \sim \frac{2}{s_u^2}\partial_x^2 s_u + \frac{3}{s_u^2} - 4\frac{(\partial_x s_u)^2}{s_u^3}, \tag{23}$$





where the symbols $\sim$ is used to indicate that this is not an equality but a proposal of closure for the term in the left-hand side,

and which leads to the closed system

$$
\begin{cases}
\partial_t u & = -u\partial_x u + \kappa\partial_x^2 u - \frac{1}{2}\partial_x V, \\
\partial_t V_u & = -u\partial_x V - 2(\partial_x u)V + \kappa\partial_x^2 V \\
& \quad -\frac{\kappa}{2}\frac{1}{V}(\partial_x V)^2 - \frac{2\kappa}{s_{u,xx}}V_u, \\
\partial_t s_{u,xx} & = -u\partial_x s_{u,xx} + 2(\partial_x u)s_{u,xx} + 4\kappa \\
& \quad -2\frac{\kappa s_{u,xx}}{V_u}\partial_x^2 V_u + 2\frac{\kappa s_{u,xx}}{V_u^2}(\partial_x V_u)^2 \\
& \quad +\kappa\frac{1}{V_u}\partial_x V_u\partial_x s_{u,xx} + \kappa\partial_x^2 s_{u,xx} \\
& \quad -2\kappa\frac{1}{s_{u,xx}}(\partial_x s_{u,xx})^2.
\end{cases}
\tag{24}
$$

The closure Eq. (23) results from a local Gaussian approximation of the correlation function. Previous numerical experiments have shown that this closure is well adapted to the Burgers' equation (Pannekoucke et al., 2018a). But the approach that has been followed to find this closure is quite specific, and it would be interesting to design a general way to find such a closure.

In particular, it would be interesting to search a generic way for designing closures that leverages on the symbolic computation, which could be plugged with the PKF dynamics computed from SymPKF at a symbolic level. To do so, we propose an empirical closure which leverages on a data-driven strategy so to hybridize machine learning with physics, as proposed by Pannekoucke and Fablet (2020) with their neural network generator `PDE-NetGen`.

The construction of the proposal relies on the symbolic computation shown in Fig. 5:

The first step is to consider an analytical approximation for the correlation function. For the illustration, we consider that the local correlation function is well approximated by the quasi-Gaussian function

$$
\rho(x, x + \delta x) \approx \exp\left(-\frac{\delta x^2}{s_u(x) + s_u(x + \delta x)}\right).
\tag{25}
$$

Then, the second step is to perform the computation of the Taylor's expansion of Eq. (22) at a symbolic level. This is done thanks to SymPy with the method `series` applied to Eq. (25) for $\delta x$ near the value $0$ and at a given order, *e.g.* for the

illustration expansion is computed as the sixth order in Fig. 5.

Then, the identification with the Taylor's expansion Eq. (22), leads to the closure

$$
\mathbb{E}\left[\varepsilon_u \partial_x^4 \varepsilon_u\right] \sim \frac{3}{s_{u,xx}^2}\partial_x^2 s_{u,xx} + \frac{3}{s_{u,xx}^2} - 3\frac{\left(\partial_x s_{u,xx}\right)^2}{s_{u,xx}^3}.
\tag{26}
$$

While it looks like the closure Eq. (23), the coefficient are not the same. But this suggests that the closure of $\mathbb{E}\left[\varepsilon_u \partial_x^4 \varepsilon_u\right]$ can be expanded as

$$
\mathbb{E}\left[\varepsilon_u \partial_x^4 \varepsilon_u\right] \sim \frac{a_0^4}{s_{u,xx}^2}\partial_x^2 s_{u,xx} + \frac{a_1^4}{s_{u,xx}^2} + a_3^4\frac{\left(\partial_x s_{u,xx}\right)^2}{s_{u,xx}^3},
\tag{27}
$$

where $\mathbf{a}^4 = (a_0^4, a_1^4, a_2^4)$ are three unknown reals. A data-driven strategy can be considered to find an appropriate value of $\mathbf{a}^4$ from experiments. This has been investigated by using the automatic generator of neural network `PDE-NetGen` which bridges





```python
import sympy
from sympkf import remove_eval_derivative, Expectation

s = pkf_burgers.fields[u].aspect[0]
eps = pkf_burgers.fields[u].epsilon
dx = sympy.Symbol('\delta x')

# definition of the correlation
rho = sympy.exp(- dx*dx / ( s + s.subs(x,x+dx) ) )

# Taylor expansion of the correlation with respect to dx, at order 6
taylor_order = 6
taylor = rho.series(dx,0,taylor_order)
taylor = remove_eval_derivative(taylor.removeO())

# Definition of pattern for the design of closure proposals
a = sympy.Wild("a",properties=[lambda k: k.is_Rational])
b = sympy.Wild("b")

for order in [4,5]:
    # Extract the term of a given order
    expr = taylor.coeff(dx,order)*sympy.factorial(order)
    if order==4: display(expr)
    # Create a proposal for the term at the order from pattern matching
    expr = sum([sympy.symbols(f'a_{k}^{order}')*term.match(a*b)[b]
                for k,term in enumerate(expr.args)] )
    display({Expectation(eps*Derivative(eps,x,order)):expr})
```

$$\frac{3\frac{\partial^2}{\partial x^2}s_{u,xx}(t,x)}{s_{u,xx}{}^2(t,x)} + \frac{3}{s_{u,xx}{}^2(t,x)} - \frac{3\left(\frac{\partial}{\partial x}s_{u,xx}(t,x)\right)^2}{s_{u,xx}{}^3(t,x)}$$

$$\left\{ \mathbb{E}\left(\varepsilon_u(t,x,\omega)\frac{\partial^4}{\partial x^4}\varepsilon_u(t,x,\omega)\right) : \frac{a_0^4}{s_{u,xx}{}^2(t,x)} + \frac{a_1^4\left(\frac{\partial}{\partial x}s_{u,xx}(t,x)\right)^2}{s_{u,xx}{}^3(t,x)} + \frac{a_2^4\frac{\partial^2}{\partial x^2}s_{u,xx}(t,x)}{s_{u,xx}{}^2(t,x)} \right\}$$

$$\left\{ \mathbb{E}\left(\varepsilon_u(t,x,\omega)\frac{\partial^5}{\partial x^5}\varepsilon_u(t,x,\omega)\right) : \frac{a_0^5\frac{\partial}{\partial x}s_{u,xx}(t,x)}{s_{u,xx}{}^3(t,x)} + \frac{a_1^5\frac{\partial^3}{\partial x^3}s_{u,xx}(t,x)}{s_{u,xx}{}^2(t,x)} + \frac{a_2^5\left(\frac{\partial}{\partial x}s_{u,xx}(t,x)\right)^3}{s_{u,xx}{}^4(t,x)} + \frac{a_3^5\frac{\partial}{\partial x}s_{u,xx}(t,x)\frac{\partial^2}{\partial x^2}s_{u,xx}(t,x)}{s_{u,xx}{}^3(t,x)} \right\}$$

**Figure 5.** Example of a symbolic computation leading to a proposal for the closure of the unknown terms of order 4 and 5.





the gap between the physics and the machine-learning (Pannekoucke and Fablet, 2020), and where the training has lead to the value $\mathbf{a}^4 \approx (0.93, 0.75, -1.80) \pm (5.1\,10^{-5}, 3.6\,10^{-4}, 2.7\,10^{-4})$ (estimation obtained from 10 runs). Since this proposal is deduced from symbolic computation, it is easy to build some proposals for higher-order unknown terms as it is shown in Fig. 5 for the term $\mathbb{E}\left[\varepsilon_u \partial_x^5 \varepsilon_u\right]$.

Whatever if the closure has been obtained from an analytical or an empirical way, it remains to compute the closed PKF dynamics to assess its performance. To do so a numerical implementation of the system of partial differential equation has to be introduced. As for the computation of the PKF dynamics, the design of a numerical code can be tedious, with a risk to introduce errors in the implementation due to the numerous terms occurring in the PKF dynamics. To facilitate the research on the PKF, SymPKF comes with a Python numerical code generator, which provides an end-to-end investigation of the PKF dynamics. This code generator is now detailed.

### 4.4 Automatic code generation for numerical simulations

While compiled language with appropriate optimization should be important for industrial applications, we chose to implement a pure Python code generator which offers a simple research framework for exploring the design of PKF dynamics. It would have been possible to use a code generator already based on SymPy (see *e.g.* Louboutin et al. (2019)) but such code generators being domain specific, it was less adapted to the investigation of the PKF for arbitrary dynamics. In place, we consider a finite difference implementation of partial derivatives with respect to spatial coordinates. The default domain to perform the computation is the periodic unit square of dimension the number of spatial coordinates. The length of the domain can be specified along each direction. The domain is regularly discretized along each direction while the number of grid-points can be specified for each direction.

The finite difference takes the form of an operator $\mathcal{F}$ that approximates any partial derivate at a second order of consistency: for any multi-index $\alpha$, $\mathcal{F}^\alpha u \underset{0}{=} \partial^\alpha u + \mathcal{O}(|\delta\mathbf{x}|^2)$ where $\mathcal{O}$ is the Landau's big O notation: for any $f$, the notation $f(\delta x) \underset{0}{=} \mathcal{O}(\delta x^2)$ means that $\lim_{\delta x \to 0} \frac{f(\delta x)}{\delta x^2}$ is finite. The operator $\mathcal{F}$ computed with respect to independent coordinates commute, *e.g.* $\mathcal{F}_{xy} = \mathcal{F}_x \circ \mathcal{F}_y = \mathcal{F}_y \circ \mathcal{F}_x$ where $\circ$ denotes the composition; but it does not commute for dependent coordinates *e.g.* $\mathcal{F}_x^2 \neq \mathcal{F}_x \circ \mathcal{F}_x$. The finite difference of partial derivative with respect to multi-index is computed sequentially *e.g.* $\mathcal{F}_{xxy} = \mathcal{F}_x^2 \circ \mathcal{F}_y = \mathcal{F}_y \circ \mathcal{F}_x^2$. The finite difference of order $\alpha$ with respect to a single spatial coordinate is the centered finite difference based on $\alpha + 1$ points.

For instance, Fig. 6 shows how to close the PKF dynamics for the Burgers' equation following P18, and how to build a code from an instance of the class `sympkf.FDModelBuilder`: it creates the class `ClosedPKFBurgers`. In this example, the code is rendered from templates thanks to Jinja[3], then it is executed at run time. Note that the code can also be written in an appropriate Python's module for adapting the code to a particular situation or to check the correctness of the generated code. At the end, the instance `closed_pkf_burgers` of the class `ClosedPKFBurgers` is created, raising a warning to indicate that the value of constant $\kappa$ has to be specified before to perform a numerical simulation. Note that it is possible to set the value of kappa as a keyword argument in the class `ClosedPKFBurgers`. Fig. 6 also show a sample of the generated code with the implementation of the computation of the first order partial derivative $\partial_x V_u$, which appears as a centered finite difference.

---

[3]https://jinja.palletsprojects.com/en/2.11.x/



```python
from sympy import Integer
from sympkf import FDModelBuilder

g = pkf_burgers.fields[u].metric[0] # metric tensor
s = pkf_burgers.fields[u].aspect[0] # aspect tensor

# loc. Gaussian closure of P18 in metric form, then in aspect form
P18_closure = Integer(3)*g**Integer(2)-Integer(2)*Derivative(g,x,2)
P18_closure = P18_closure.subs(g,1/s).doit().expand()

# Introduction of the closure as a dictionnary
unclosed_term = list(pkf_burgers.unclosed_terms)[0]
pkf_burgers.set_closure({unclosed_term:P18_closure})

# Build a numerical code at runtime
exec(FDModelBuilder(pkf_burgers.in_aspect, class_name='ClosedPKFBurgers').code)
closed_pkf_burgers = ClosedPKFBurgers(shape=(241,))
```

Warning: constant `kappa` has to be set

Sample of the numerical code generated in the class ClosedPKFBurgers

```python
[..]
# Compute derivatives
#-----------------------
DV_u_x_o1 = (-V_u[np.ix_(self.index('x',-1))] +
    V_u[np.ix_(self.index('x',1))])/(2*self.dx[self.coordinates.index('x')])

[..]
# Implementation of the trend
#-------------------------------
du[:] = -DV_u_x_o1/2 - Du_x_o1*u + Du_x_o2*kappa

dV_u[:] = -DV_u_x_o1**2*kappa/(2*V_u) - DV_u_x_o1*u + DV_u_x_o2*kappa -
    2*Du_x_o1*V_u - 2*V_u*kappa/s_u_xx

ds_u_xx[:] = 2*DV_u_x_o1**2*s_u_xx*kappa/V_u**2 +
    DV_u_x_o1*Ds_u_xx_x_o1*kappa/V_u -
    2*DV_u_x_o2*s_u_xx*kappa/V_u - 2*Ds_u_xx_x_o1**2*kappa/s_u_xx -
    Ds_u_xx_x_o1*u +
    Ds_u_xx_x_o2*kappa + 2*Du_x_o1*s_u_xx + 4*kappa
[..]
```

**Figure 6.** Introduction of a closure and automatic generation of a numerical code in SymPKF.





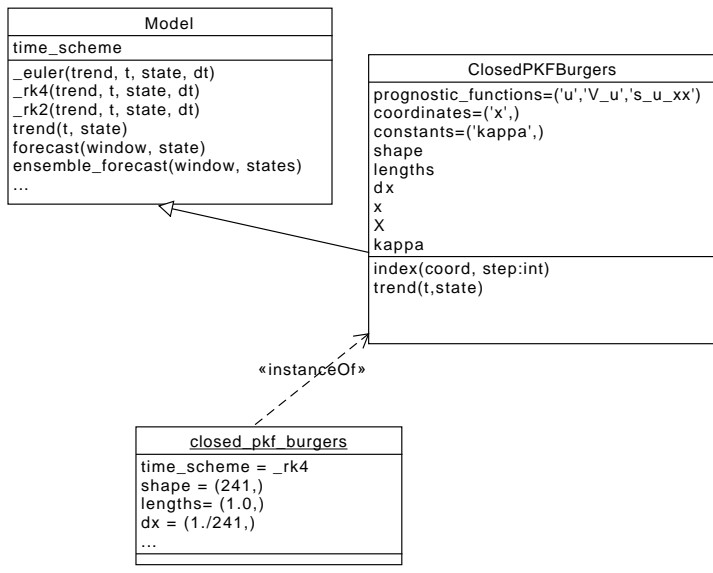

**Figure 7.** UML diagram showing the inheritance mechanism implemented in SymPKF: the class `ClosedPKFBurgers` inherits from the class `Model` which implements several time schemes. Here, `closed_pkf_burgers` is an instance of the class `ClosedPKFBurgers`.

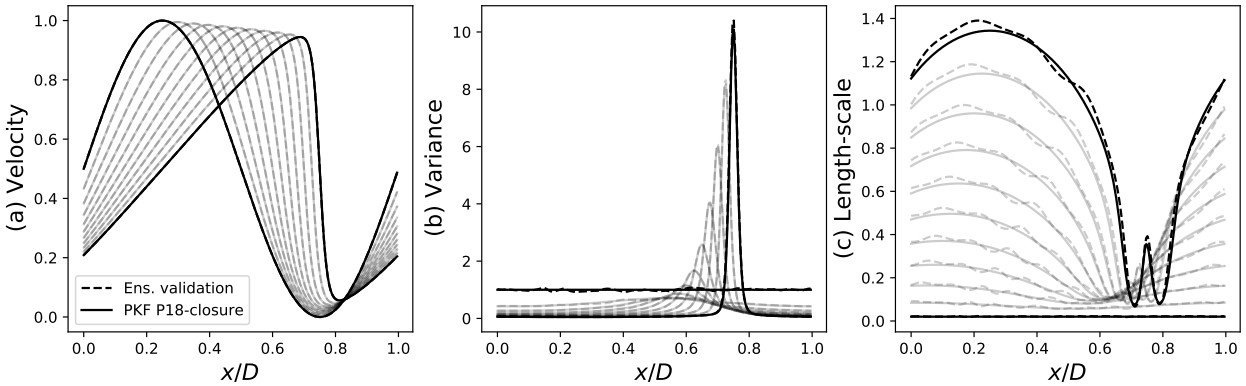

**Figure 8.** Illustration of a numerical simulation of the PKF dynamics Eq. (23) (solid line), with the mean (panel a), the variance (panel b) and the correlation length-scale (panel c) which is defined, from the component $s_{u,xx}$ of the aspect tensor, by $L(x) = \sqrt{s_{u,xx}(x)}$. An ensemble-based validation of the PKF dynamics is shown in dashed line. (Pannekoucke and Fablet, 2020, see their Fig. 7)


Then, the sample of code shows how the partial derivatives are used to compute the trend of the system of partial differential equations Eq. (24).

The numerical integration is handled through the inheritance mechanism: the class `ClosedPKFBurgers` inherits the integration time loop from the class `sympkf.Model` as described by the UML diagram shown in Fig. 7. In particular, the class

`Model` contains several time schemes *e.g.* a fourth-order Runge-Kutta scheme. The details of the instance `closed_pkf_burgers` of the class `ClosedPKFBurgers` make appear that the closed system Eq. (24) will be integrated by using a RK4 time scheme, on the segment $[0, D]$ (here $D = 1$) with periodic boundaries, and discretized by 241 points.

Thanks to the end-to-end framework proposed in SymPKF, it is possible to perform a numerical simulation based on the PKF dynamics Eq. (23). To do so, we set $\kappa = 0.0025$ and consider the simulation starting from the Gaussian distribution $\mathcal{N}(u_0, \mathbf{P}_h^f)$

of mean $u_0(x) = U_{max}[1 + \cos(2\pi(x - D/4)/D)]/2$ with $U_{max} = 0.5$, and of covariance matrix

$$\mathbf{P}_h^f(x, y) = V_h \exp\left(-\frac{(x - y)^2}{2l_h^2}\right), \tag{28}$$

where $V_h = 0.01 U_{max}$ and $l_h = 0.02D \approx 5dx$. The time step of the fourth-order Runge-Kutta scheme is $dt = 0.002$. The evolution predicted from the PKF is shown in Fig. 8 (solid lines). This simulation illustrates the time evolution of the mean (panel a) and of the variance (panel b) ; the panel (c) represents the evolution of the correlation length-scale defined from the

aspect tensor as $L(x) = \sqrt{s_{u,xx}(x)}$. Note that at time 0, the length-scale field is $L(x) = l_h$. For the illustrations, the variance (the length-scale) is normalized by its initial value $V_h$ ($l_h$).

In order to show the skill of the PKF applied on the Burgers' equation, when using the closure of P18, an ensemble validation is now performed. Note that the code generator of SymPKF can be used for an arbitrary dynamics *e.g.* the Burgers' equation itself. Hence, a numerical code solving the Burgers' equation is rendered from its symbolic definition. Then an ensemble of

1600 forecasts is computed starting from an ensemble of initial errors at time 0. The ensemble of initial errors is sampled from the Gaussian distribution $\mathcal{N}\left(0, \mathbf{P}_h^f\right)$ of zero mean and covariance matrix $\mathbf{P}_h^f$. Note that the ensemble forecasting implemented in SymPKF as the method `Model.ensemble_forecast` (see Fig. 7) leverages on the multiprocessing tools of Python, so to use the multiple cores of the CPU, when present. On the computer used for the simulation, the forecasts are performed in parallel on the 8 cores. The ensemble estimation of the mean, the variance and the length-scale are shown in Fig. 8 (dashed

lines). Since the ensemble is finite, a sampling noise is visible *e.g.* on the variance at the initial time that is not strictly equal to $V_h$. In this simulation, it appears that the PKF (solid line) coincide with the ensemble estimation (dashed lines) which shows the ability of the PKF to predict the forecast-error covariance dynamics. Note that the notebook corresponding to the Burgers' experiment is available in the example directory of SymPKF.

While this example shows an illustration of SymPKF in 1D domain, the package also applies in 2D and 3D domains, as

presented now.





```
from sympkf.symbolic import SymbolicPKF
pkf_advection = SymbolicPKF(dynamics)
```

```
for equation in pkf_advection.in_metric:    display(equation)
```

$$\frac{\partial}{\partial t}c(t,x,y) = u(x,y)\frac{\partial}{\partial x}c(t,x,y) + v(x,y)\frac{\partial}{\partial y}c(t,x,y)$$

$$\frac{\partial}{\partial t}V_c(t,x,y) = u(x,y)\frac{\partial}{\partial x}V_c(t,x,y) + v(x,y)\frac{\partial}{\partial y}V_c(t,x,y)$$

$$\frac{\partial}{\partial t}g_{c,xx}(t,x,y) = u(x,y)\frac{\partial}{\partial x}g_{c,xx}(t,x,y) + v(x,y)\frac{\partial}{\partial y}g_{c,xx}(t,x,y) + 2g_{c,xx}(t,x,y)\frac{\partial}{\partial x}u(x,y) + 2g_{c,xy}(t,x,y)\frac{\partial}{\partial x}v(x,y)$$

$$\frac{\partial}{\partial t}g_{c,xy}(t,x,y) = u(x,y)\frac{\partial}{\partial x}g_{c,xy}(t,x,y) + v(x,y)\frac{\partial}{\partial y}g_{c,xy}(t,x,y) + g_{c,xx}(t,x,y)\frac{\partial}{\partial y}u(x,y) + g_{c,xy}(t,x,y)\frac{\partial}{\partial x}u(x,y) + g_{c,xy}(t,x,y)\frac{\partial}{\partial y}v(x,y) + g_{c,yy}(t,x,y)\frac{\partial}{\partial x}v(x,y)$$

$$\frac{\partial}{\partial t}g_{c,yy}(t,x,y) = u(x,y)\frac{\partial}{\partial x}g_{c,yy}(t,x,y) + v(x,y)\frac{\partial}{\partial y}g_{c,yy}(t,x,y) + 2g_{c,xy}(t,x,y)\frac{\partial}{\partial y}u(x,y) + 2g_{c,yy}(t,x,y)\frac{\partial}{\partial y}v(x,y)$$

```
for equation in pkf_advection.in_aspect:    display(equation)
```

$$\frac{\partial}{\partial t}c(t,x,y) = u(x,y)\frac{\partial}{\partial x}c(t,x,y) + v(x,y)\frac{\partial}{\partial y}c(t,x,y)$$

$$\frac{\partial}{\partial t}V_c(t,x,y) = u(x,y)\frac{\partial}{\partial x}V_c(t,x,y) + v(x,y)\frac{\partial}{\partial y}V_c(t,x,y)$$

$$\frac{\partial}{\partial t}s_{c,xx}(t,x,y) = u(x,y)\frac{\partial}{\partial x}s_{c,xx}(t,x,y) + v(x,y)\frac{\partial}{\partial y}s_{c,xx}(t,x,y) - 2s_{c,xx}(t,x,y)\frac{\partial}{\partial x}u(x,y) - 2s_{c,xy}(t,x,y)\frac{\partial}{\partial y}u(x,y)$$

$$\frac{\partial}{\partial t}s_{c,xy}(t,x,y) = u(x,y)\frac{\partial}{\partial x}s_{c,xy}(t,x,y) + v(x,y)\frac{\partial}{\partial y}s_{c,xy}(t,x,y) - s_{c,xx}(t,x,y)\frac{\partial}{\partial x}v(x,y) - s_{c,xy}(t,x,y)\frac{\partial}{\partial x}u(x,y) - s_{c,xy}(t,x,y)\frac{\partial}{\partial y}v(x,y) - s_{c,yy}(t,x,y)\frac{\partial}{\partial y}u(x,y)$$

$$\frac{\partial}{\partial t}s_{c,yy}(t,x,y) = u(x,y)\frac{\partial}{\partial x}s_{c,yy}(t,x,y) + v(x,y)\frac{\partial}{\partial y}s_{c,yy}(t,x,y) - 2s_{c,xy}(t,x,y)\frac{\partial}{\partial x}v(x,y) - 2s_{c,yy}(t,x,y)\frac{\partial}{\partial y}v(x,y)$$

**Figure 9.** Sample of code and Jupyter notebook outputs: system of partial differential equations produced by SymPKF when applied to the linear advection Eq. (29).





## 4.5 Illustration of a dynamics in a 2D domain

In order to illustrate the ability of SymPKF to apply in a 2D or a 3D domain, we consider the linear advection of a scalar field $c(t,x,y)$ by a stationary velocity field $\mathbf{u} = (u(x,y), v(x,y))$, which reads as the partial differential equation

$$\partial_t c + \mathbf{u}\nabla c = 0. \tag{29}$$

As for the Burgers' equation, the definition of the dynamics relies on SymPy (not shown but similar to the definition of the Burgers' equation as given in Fig. 1). This leads to preprocessing the dynamics by creating the instance *advection* of the class *PDESystem*, which transform the equation into a system of partial differential equations. In particular, the procedure will diagnose the prognostic functions of a dynamics, here the function $c$ ; the constant functions, here $u, v$ the component of the velocity $(u,v)$ ; for this example there is no constant nor exogenous function.

The calculation of the parametric dynamics is handled by the class *SymbolicPKF* as shown in the first cell in Fig. 9. The parametric dynamics is a property of the instance *pkf_advection* of the class *SymbolicPKF*, and when it is called, the parametric dynamics is computed once and for all. The parametric dynamics formulated in terms of metric is first computed, see the second cell. For the 2D linear advection, the parametric dynamics is a system of five partial differential equations, as it is shown in the output of the second cell: the dynamics of the ensemble average $\mathbb{E}[c]$ which outputs as $c$ for the sake of simplicity (first equation), the dynamics of the variance (second equation) and the dynamics of the local metric tensor (last three equations). In compact form, the dynamics is given by the system

$$\partial_t c + \mathbf{u}\nabla c = 0, \tag{30a}$$

$$\partial_t V_c + \mathbf{u}\nabla V_c = 0, \tag{30b}$$

$$\partial_t \mathbf{g}_c + \mathbf{u}\nabla \mathbf{g}_c = -\mathbf{g}_c (\nabla \mathbf{u}) - (\nabla \mathbf{u})^T \mathbf{g}_c, \tag{30c}$$

which corresponds to the 2D extension of the 1D dynamics first found by Cohn (1993) (Pannekoucke et al., 2016), and validates the computation performed in SymPKF. Due to the linearity of the linear advection Eq. (29), the ensemble average Eq. (30a) is governed by the same dynamics Eq. (29). While both the variance, Eq. (30b), and the metric are advected by the flow, the metric is also deformed by the shear Eq. (30c). This deformation more commonly appears on the dynamics written in aspect tensor form, which is given by

$$\partial_t c + \mathbf{u}\nabla c = 0, \tag{31a}$$

$$\partial_t V_c + \mathbf{u}\nabla V_c = 0, \tag{31b}$$

$$\partial_t \mathbf{s}_c + \mathbf{u}\nabla \mathbf{s}_c = (\nabla \mathbf{u}) \mathbf{s}_c + \mathbf{s}_c (\nabla \mathbf{u})^T, \tag{31c}$$

where Eq. (31c) is similar to the dynamics of the conformation tensor in viscoelastic flow (Bird and Wiest, 1995; Hameduddin et al., 2018).

We do not introduce any numerical simulation of the PKF dynamics Eq. (30) or Eq. (31), but the interested readers are referred to the 2D numerical PKF assimilation cycles of Pannekoucke (2021b), which have been made thanks to SymPKF.





This example illustrates a 2D situation and shows the multidimensional capabilities of SymPKF. Similarly to the simulation conducted for the Burgers' equation, it is possible to automatically generate a numerical code able to perform numerical simulations of the dynamics Eq. (31) (not shown here). Hence, this 2D domain example showed the ability of SymPKF to apply in dimensions lager than the 1D.

Before concluding, we would like to present a preliminary application of SymPKF in a multivariate situation.

### 4.6 Towards the PKF for multivariate dynamics

SymPKF can be used to compute the prediction of the variance and the anisotropy in a multivariate situation.

Note that one of the difficulty with the multivariate situation is that the number of equations increases linearly with the number of fields and the dimension of the domain *e.g.* for a 1D (2D) domain and two multivariate physical fields, there are two ensemble averaged fields, two variance fields and two (six) metric fields. Of course this is no not a problem when using a computer algebra system as done in SymPKF.

So to illustrate the multivariate situation, only a very simple example is introduced. Inspired from chemical transport models encountered in air quality, we consider the transport, over a 1D domain, of two chemical species, whose concentrations are denoted by $A(t,x)$ and $(B(t,x)$, and advected by the wind $u(x)$. For the sake of simplicity, the two species interact following a periodic dynamics as defined by the coupled system

$$\partial_t A + u \partial_x A = B, \tag{32a}$$
$$\partial_t B + u \partial_x B = -A, \tag{32b}$$

Thanks to the splitting strategy, the PKF dynamics due to the advection has already been detailed in the previous section (see Section 4.5), so we can focus on the chemical part of the dynamics which is given by the processes on the right-hand side of Eq. (32). The PKF of the chemical part is computed thanks to SymPKF, and shown in Fig. 10. This time, and as it is expected, multivariate statistics appear in the dynamics. Here, the dynamics of the cross-covariance $V_{AB} = \mathbb{E}[e_A e_B]$ is given by the fifth equation. The coupling makes appear unknown terms *e.g.* the term $\mathbb{E}[\partial_x \varepsilon_A \partial_x \varepsilon_B]$ in the sixth equation of the output shown in Fig. 10.

To go further, some research is still needed to explore the dynamics and the modelling of the multivariate cross covariances. A possible direction is to take advantage of the multivariate covariance model based on balance operator as often introduced in variational data assimilation (Derber and Bouttier, 1999; Ricci et al., 2005). Note that such multivariate covariance models has been recently considered for the design of the multivariate PKF analysis step (Pannekoucke, 2021b). Another way is to consider a data-driven strategy to learn the physics of the unknown terms, from a training based on ensembles of forecasts (Pannekoucke and Fablet, 2020).

To conclude, this example shows the potential of interest of SymPKF to tackle the multivariate situation. Moreover, the example also shows that SymPKF is able to perform the PKF computation for a system of partial differential equations. However, all the equations should be prognostic, SymPKF is not able to handle diagnostic equations.





```
A = Function('A')(t,x)
B = Function('B')(t,x)
dynamics = [Eq(Derivative(A,t), B), Eq(Derivative(B,t), -A)]
for equation in SymbolicPKF(dynamics).in_metric:    display(equation)
```

$$\frac{\partial}{\partial t} A(t,x) = B(t,x)$$

$$\frac{\partial}{\partial t} B(t,x) = -A(t,x)$$

$$\frac{\partial}{\partial t} V_A(t,x) = 2 V_{AB}(t,x)$$

$$\frac{\partial}{\partial t} V_B(t,x) = -2 V_{AB}(t,x)$$

$$\frac{\partial}{\partial t} V_{AB}(t,x) = -V_A(t,x) + V_B(t,x)$$

$$\frac{\partial}{\partial t} g_{A,xx}(t,x) = -\frac{2 V_{AB}(t,x) g_{A,xx}(t,x)}{V_A(t,x)} + \frac{2\sqrt{V_B(t,x)}\mathbb{E}\left(\frac{\partial}{\partial x}\varepsilon_A(t,x,\omega)\frac{\partial}{\partial x}\varepsilon_B(t,x,\omega)\right)}{\sqrt{V_A(t,x)}} +$$
$$\frac{\mathbb{E}\left(\varepsilon_B(t,x,\omega)\frac{\partial}{\partial x}\varepsilon_A(t,x,\omega)\right)\frac{\partial}{\partial x}V_B(t,x)}{\sqrt{V_A(t,x)}\sqrt{V_B(t,x)}} - \frac{\sqrt{V_B(t,x)}\mathbb{E}\left(\varepsilon_B(t,x,\omega)\frac{\partial}{\partial x}\varepsilon_A(t,x,\omega)\right)\frac{\partial}{\partial x}V_A(t,x)}{V_A^{\frac{3}{2}}(t,x)}$$

$$\frac{\partial}{\partial t} g_{B,xx}(t,x) = \frac{2 V_{AB}(t,x) g_{B,xx}(t,x)}{V_B(t,x)} - \frac{2\sqrt{V_A(t,x)}\mathbb{E}\left(\frac{\partial}{\partial x}\varepsilon_A(t,x,\omega)\frac{\partial}{\partial x}\varepsilon_B(t,x,\omega)\right)}{\sqrt{V_B(t,x)}} +$$
$$\frac{\sqrt{V_A(t,x)}\mathbb{E}\left(\varepsilon_A(t,x,\omega)\frac{\partial}{\partial x}\varepsilon_B(t,x,\omega)\right)\frac{\partial}{\partial x}V_B(t,x)}{V_B^{\frac{3}{2}}(t,x)} - \frac{\mathbb{E}\left(\varepsilon_A(t,x,\omega)\frac{\partial}{\partial x}\varepsilon_B(t,x,\omega)\right)\frac{\partial}{\partial x}V_A(t,x)}{\sqrt{V_A(t,x)}\sqrt{V_B(t,x)}}$$

**Figure 10.** Output of the computation by SymPKF of the PKF dynamics for the simple multivariate periodic chemical reaction, corresponding to the right-hand side of Eq. (32).

## 5   Conclusions

This contribution introduced the package SymPKF that can be used to conduct the research on the parametric Kalman filter prediction step, for covariance models parameterized by the variance and the anisotropy (VLATcov models). SymPKF provides an end-to-end framework: from the equations of a dynamics to the development of a numerical code.

The package has been first introduced by considering the non-linear diffusive advection dynamics, the Burgers' equation. In particular this example shows the ability of SymPKF to handle abstract terms *e.g.* the unclosed terms formulated with the

expectation operator. The expectation operator implemented in SymPKF is a key tool for the computation of the PKF dynamics. Moreover, we showed how to handle a closure and how to automatically render numerical codes.

For univariate situations, SymPKF applies in 1D domain as well as in 2D and 3D domains. This has been shown by considering the computation of the PKF dynamics for the linear advection equation on a 2D domain.





A preliminary illustration on a multivariate dynamics showed the potential of SymPKF to handle the dynamics of multivari-
ate covariance. But this point has to be further investigated, and this constitutes the main perspective of development. Moreover,
to perform a multivariate assimilation cycle with the PKF, the multivariate formulation of the PKF analysis state is needed. A
first investigation of the multivariate PKF assimilation has been proposed by Pannekoucke (2021b).

In its present implementation, SymPKF is limited to the computation with prognostic equations. It is not possible to consider
dynamics based on diagnostic equations while these are often encountered in atmospheric fluid dynamics *e.g.* the geostrophic
balance. This constitutes another topic of research development for the PKF, facilitated by the use of symbolic exploration.

Note that the expectation operator as introduced here can be used to compute Reynolds equations encountered in turbulence.
This open new perspectives of use of SymPKF for other applications that could be interesting especially for automatic code
generation.

*Code and data availability.* The SymPKF package is free and open source. It is distributed under the CeCILL-B free software licence. The
source code is provided through a GitHub repository at https://github.com/opannekoucke/sympkf, last access: 22 Mars 2021. A snapshot of
SymPKF is available at https://doi.org/10.5281/zenodo.4608514 (Pannekoucke, 2021c). The data used for the simulations presented here are
generated at runtime when using the Jupyter notebooks.

## Appendix A: Splitting for the computation of the parametric dynamics

In this section we show that using a splitting strategy is possible for the design of the parametric dynamics. For this, it is enough
to show that given a dynamics written as

$$\partial_t \mathcal{X} = f_1(\mathcal{X}) + f_2(\mathcal{X}), \tag{A1}$$

the dynamics of the error, the variance, the metric and the aspect all write as a sum of trends depending on each process $f_1$ and
$f_2$. We show this starting from the dynamics of the error.

Due to the linearity of the derivative operator, the TL dynamics resulting from Eq. (A1) writes

$$\partial_t e = f_1'(e) + f_2'(e), \tag{A2}$$

where $f_1'$ and $f_2'$ denote the differential of the two functions. which can be written as the sum of two trends $\partial_t e_1 = f_1'(e)$ and
$\partial_t e_2 = f_2'(e)$, depending exclusively on $f_1$ and $f_2$ respectively. For the variance's dynamics, $\partial_t V = 2\mathbb{E}[e\partial_t e]$, substitution by
Eq. (A2) leads to

$$\partial_t V = \partial_t V_1 + \partial_t V_2, \tag{A3}$$

where $\partial_t V_1 = 2\mathbb{E}[ef_1'(e)]$ and $\partial_t V_2 = 2\mathbb{E}[ef_2'(e)]$, depends exclusively on $f_1$ and $f_2$ respectively. Then the standard deviation
dynamics, obtained by differentiating $\sigma^2 = V$ as $2\sigma\partial_t\sigma = \partial_t V$,

$$\partial_t \sigma = \frac{1}{\sigma}\partial_t V_1 + \frac{1}{\sigma}\partial_t V_2, \tag{A4}$$





reads as the sum of two trends $\partial_t\sigma_1 = \frac{1}{\sigma}\partial_t V_1$ and $\partial_t\sigma_2 = \frac{1}{\sigma}\partial_t V_2$, depending exclusively on $f_1$ and $f_2$ respectively. It results that the dynamics of the normalized error $\varepsilon = \frac{1}{\sigma}e$, deduced from the time derivative of $e = \sigma\varepsilon$, $\partial_t e = \varepsilon\partial_t\sigma + \sigma\partial_t\varepsilon$, reads as

$$\partial_t\varepsilon = \frac{1}{\sigma}\left[f_1'(e) - \frac{\varepsilon}{2\sigma}\partial_t V_1\right] + \frac{1}{\sigma}\left[f_2'(e) - \frac{\varepsilon}{2\sigma}\partial_t V_2\right] \tag{A5}$$

and also expands as the sum of two trends $\partial_t\varepsilon_1 = \frac{1}{\sigma}\left[f_1'(e) - \frac{\varepsilon}{2\sigma}\partial_t V_1\right]$ and $\partial_t\varepsilon_2 = \frac{1}{\sigma}\left[f_2'(e) - \frac{\varepsilon}{2\sigma}\partial_t V_2\right]$, again depending exclusively on $f_1$ and $f_2$ respectively. For the metric terms $g_{ij} = \mathbb{E}\left[\partial_i\varepsilon\partial_j\varepsilon\right]$, we deduce that the dynamics $\partial_t g_{ij} = \mathbb{E}\left[\partial_i(\partial_t\varepsilon)\partial_j\varepsilon\right] + \mathbb{E}\left[\partial_i\varepsilon\partial_j(\partial_t\varepsilon)\right]$ expands as

$$\partial_t g_{ij} = \partial_t g_{ij_1} + \partial_t g_{ij_2}, \tag{A6}$$

with $\partial_t g_{ij_1} = \mathbb{E}\left[\partial_i(\partial_t\varepsilon_1)\partial_j\varepsilon\right] + \mathbb{E}\left[\partial_i\varepsilon\partial_j(\partial_t\varepsilon_1)\right]$ and $\partial_t g_{ij_2} = \mathbb{E}\left[\partial_i(\partial_t\varepsilon_2)\partial_j\varepsilon\right] + \mathbb{E}\left[\partial_i\varepsilon\partial_j(\partial_t\varepsilon_2)\right]$ where each partial trend depends exclusively on $f_1$ and $f_2$ respectively. To end, dynamics of the aspect tensor $\mathbf{s}$ is deduced from Eq. (17) which expands as

$$\partial_t\mathbf{s} = \partial_t\mathbf{s}_1 + \partial_t\mathbf{s}_2, \tag{A7}$$

where $\partial_t\mathbf{s}_1 = -\mathbf{s}(\partial_t\mathbf{g}_1)\mathbf{s}$ and $\partial_t\mathbf{s}_2 = -\mathbf{s}(\partial_t\mathbf{g}_2)\mathbf{s}$ only depend of on $f_1$ and $f_2$ respectively.

To conclude, the computation of the parametric dynamics for Eq. (A1), can be performed from the parametric dynamics of $\partial_t\mathcal{X} = f_1(\mathcal{X})$ and $\partial_t\mathcal{X} = f_2(\mathcal{X})$ calculated separately, then merged together to obtain the dynamics of the variance Eq. (A3), of the metric Eq. (A6) and of the aspect Eq. (A7) tensors.

## Appendix B: Computation of terms $\mathbb{E}\left[\partial^\alpha\varepsilon\partial^\beta\varepsilon\right]$

In this section we proof the property

**Property 1.** *Any term* $\mathbb{E}\left[\partial^\alpha\varepsilon\partial^\beta\varepsilon\right]$ *with* $|\alpha| \leq |\beta|$*, can be related to the correlation expansion terms* $\mathbb{E}\left[\varepsilon\partial^\gamma\varepsilon\right]$ *where* $|\gamma| < |\alpha| + |\beta|$*, and the term* $\mathbb{E}\left[\varepsilon\partial^{\alpha+\beta}\varepsilon\right]$.

**Proof:**

The derivative with respect to a zero $\alpha_i$ is the identity operator. Note that the multi-index forms a semi-group since for two multi-index $\alpha$ and $\beta$ we can form the multi-index $\alpha + \beta = (\alpha_i + \beta_i)_{i\in[1,n]}$.

Now the property Eq. (1) can be proven considering the following recurrent process, when asuming that the property is true for all patterns of degree strictly lower to the degree $|\alpha| + |\beta|$:

Without loss of generality we assume $\alpha_i > 0$ and denote $\delta_i = (\delta_{ij})_{j\in[1,n]}$ where $\delta_{ij}$ is the Kroenecker symbol ($\delta_{ii} = 1$, $\delta_{ij} = 0$ for $j \neq i$). From the formula

$$\partial_{x^i}\left(\partial^{\alpha-\delta_i}\varepsilon\partial^\beta\varepsilon\right) = \partial^\alpha\varepsilon\partial^\beta\varepsilon + \partial^{\alpha-\delta_i}\varepsilon\partial^{\beta+\delta_i}\varepsilon \tag{B1}$$

and from the commutativity of the expectation operator and the partial derivative with respect to the coordinate system, it results that

$$\mathbb{E}\left[\partial^\alpha\varepsilon\partial^\beta\varepsilon\right] = \partial_{x^i}\mathbb{E}\left[\partial^{\alpha-\delta_i}\varepsilon\partial^\beta\varepsilon\right] - \mathbb{E}\left[\partial^{\alpha-\delta_i}\varepsilon\partial^{\beta+\delta_i}\varepsilon\right]. \tag{B2}$$





Considering the terms of the left-hand side. In one hand, we observe that the degree of the first term is decreasing to $|\alpha|+|\beta|-1$, from the recurrence assumption $\mathbb{E}\left[\partial^{\alpha-\delta_i}\varepsilon\partial^\beta\varepsilon\right]$ can be expanded as terms of the form $\mathbb{E}\left[\varepsilon\partial^\gamma\varepsilon\right]$. On the other hand, the degree

of the second term remains the same, $|\alpha|+|\beta|$, but with a shift of the derivative order. This shift of the order can be done again following the same process, leading after iterations to the term $\mathbb{E}\left[\varepsilon\partial^{\alpha+\beta}\varepsilon\right]$.

*Author contributions.* OP introduced the symbolic computation of the PKF dynamics, OP and PA imagined an end-to-end framework for the design of the PKF dynamics: from the equation of the dynamics to the numerical simulation thanks to an automatic code generation. OP developed the codes.

*Competing interests.* The authors declare that they have no conflict of interest.

*Acknowledgements.* The UML class diagram has been generated from UMLlet (Auer et al., 2003). This work was supported by the French national programme LEFE/INSU (Étude du filtre de KAlman PAramétrique, KAPA).





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
