# Peer review of "SymPKF (v1.0): a symbolic and computational toolbox for the design of parametric Kalman filter dynamics"

_Geoscientific Model Development, 2021_

## Author Comment (AC1)

First of all, we would like to thank the referee for his/her review on our paper and for giving us the opportunity to improve our paper. We added our acknowledgements to the referee in the new manuscript.

Now, we organized the answer to the comments as follows. First, we list some changes afford to the manuscript then detail our answers to the questions raised by the referee.

**List of changes for the revision**

*Minor changes*

There was an error of sign in the definition of the transport dynamics shown in Fig 9, where the trend should have been defined as minus the velocity times the gradient of concentration. This has been corrected.

*Differences between the two versions of the manuscript*

To facilitate the comparison between the two version of the manuscript, a companion version of the manuscript lists all the modifications where old (new) statements are in red (blue). But the line numbers will refer to the revised version of the manuscript (not to the companion version).

**Answer to the question of the referees**

We copied your commentary in italics below, we reply in normal blue font.

General feedback:

*1. "Section 2.3 seems light on description of a fundamental part of the paper. The section above seems to allude to there being a significant savings in terms of memory for a parametrized covariance approach, though this is not explicitly shown in this section, nor is there any discussion about possible parametrizations, other possible benefits, and the disadvantages of such an approach (errors, potential non-physical covariances, etc.). A comparison to other covariance approximation methods in filtering (low-rank methods, the ensemble Kalman filter, etc) would also be worth-while."*

We extended the section 2.3 so to address the several points mentioned by the referee:
- we show how the choice of a covariance model leads to the reduction of the numerical cost,
- we compare the PKF with low-rank methods as the reduced rank Kalman filter and the EnKF,
- we introduce the need to an appropriate covariance model depending on the flow-dependency as it is encountered in geophysics, which underline the potential non-physical covariances e.g. when considering the covariance model based on the diagonal assumption in spectral space.
See l125-l152.

*2. Figure 1 seems to be cut off on the right (the bounding box is not fully shown like in some of the*

*other code figures).*

The Figures have been corrected. Thank you.

*3. Figure 2, unlike figure 1, does not have comments in the code, which is a slight inconsistency. Same in figure 4.*

We added comments in the figures where they were missing.

*4. In figure 7, one of the time integration schemes listed seems to be forward Euler. While it seems trivial to implement, for many possible equations, an implicit method could be prefered. As the authors are already dealing with symbolic schemes, a comment about possible symbolic derivations of higher order derivatives needed for such a method would be appreciated in the text.*

We introduced a comment about the use of symbolic computation for the design of other times schemes (l417):
"Note that in the present implementation of SymPKF, only explicit time schemes are considered, but it could be possible to leverage on the symbolic computation to implement other schemes more adapted to a given PDE e.g. an implicit scheme for the transport or the diffusion, or a high order exponential time-differencing method (Kassam and Trefethen, 2005) where the linear and the non-linear part would be automatically determined from the symbolic computation."

*5. In general the figures seem to be slighly inconsistent in terms of the fonts used and the sizes of the text. I would appreciate the authors double checking all the figures for such things.*

The inconsistency have been corrected. Thank you.

*6. I greatly appreciate the discussion about the limitations not just of the package, but symbolic computation in general in the conclusion.*

Thank you.

[revised manuscript text omitted]
}\,\mathrm{g_{c,xx}}\,(t,x,y) \quad = \quad -u(x,y)\frac{\partial}{\partial x}\,\mathrm{g_{c,xx}}\,(t,x,y) \;-\; v(x,y)\frac{\partial}{\partial y}\,\mathrm{g_{c,xx}}\,(t,x,y) \;-\; 2\,\mathrm{g_{c,xx}}\,(t,x,y)\frac{\partial}{\partial x}u(x,y) \;-\; 2\,\mathrm{g_{c,xy}}\,(t,x,y)\frac{\partial}{\partial x}v(x,y)$$

$$\frac{\partial}{\partial t}\,\mathrm{g_{c,xy}}\,(t,x,y) \quad = \quad -u(x,y)\frac{\partial}{\partial x}\,\mathrm{g_{c,xy}}\,(t,x,y) \;-\; v(x,y)\frac{\partial}{\partial y}\,\mathrm{g_{c,xy}}\,(t,x,y) \;-\; \mathrm{g_{c,xx}}\,(t,x,y)\frac{\partial}{\partial y}u(x,y) \;-\; \mathrm{g_{c,xy}}\,(t,x,y)\frac{\partial}{\partial x}u(x,y) - \mathrm{g_{c,xy}}\,(t,x,y)\frac{\partial}{\partial y}v(x,y) - \mathrm{g_{c,yy}}\,(t,x,y)\frac{\partial}{\partial x}v(x,y)$$

$$\frac{\partial}{\partial t}\,\mathrm{g_{c,yy}}\,(t,x,y) \quad = \quad -u(x,y)\frac{\partial}{\partial x}\,\mathrm{g_{c,yy}}\,(t,x,y) \;-\; v(x,y)\frac{\partial}{\partial y}\,\mathrm{g_{c,yy}}\,(t,x,y) \;-\; 2\,\mathrm{g_{c,xy}}\,(t,x,y)\frac{\partial}{\partial y}u(x,y) \;-\; 2\,\mathrm{g_{c,yy}}\,(t,x,y)\frac{\partial}{\partial y}v(x,y)$$

```
**Compute the PKF system rendered in aspect tensor form (the computatation is only␣**
 ↪performed at the first call)
for equation in pkf_advection.in_aspect:    display(equation)
```

$$\frac{\partial}{\partial t}c(t,x,y) = -u(x,y)\frac{\partial}{\partial x}c(t,x,y) - v(x,y)\frac{\partial}{\partial y}c(t,x,y)$$

$$\frac{\partial}{\partial t}\,\mathrm{V_c}\,(t,x,y) = -u(x,y)\frac{\partial}{\partial x}\,\mathrm{V_c}\,(t,x,y) - v(x,y)\frac{\partial}{\partial y}\,\mathrm{V_c}\,(t,x,y)$$

$$\frac{\partial}{\partial t}\,\mathrm{s_{c,xx}}\,(t,x,y) \quad = \quad -u(x,y)\frac{\partial}{\partial x}\,\mathrm{s_{c,xx}}\,(t,x,y) \;-\; v(x,y)\frac{\partial}{\partial y}\,\mathrm{s_{c,xx}}\,(t,x,y) \;+\; 2\,\mathrm{s_{c,xx}}\,(t,x,y)\frac{\partial}{\partial x}u(x,y) \;+\; 2\,\mathrm{s_{c,xy}}\,(t,x,y)\frac{\partial}{\partial y}u(x,y)$$

$$\frac{\partial}{\partial t}\,\mathrm{s_{c,xy}}\,(t,x,y) \quad = \quad -u(x,y)\frac{\partial}{\partial x}\,\mathrm{s_{c,xy}}\,(t,x,y) \;-\; v(x,y)\frac{\partial}{\partial y}\,\mathrm{s_{c,xy}}\,(t,x,y) \;+\; \mathrm{s_{c,xx}}\,(t,x,y)\frac{\partial}{\partial x}v(x,y) \;+\; \mathrm{s_{c,xy}}\,(t,x,y)\frac{\partial}{\partial x}u(x,y) + \mathrm{s_{c,xy}}\,(t,x,y)\frac{\partial}{\partial y}v(x,y) + \mathrm{s_{c,yy}}\,(t,x,y)\frac{\partial}{\partial y}u(x,y)$$

$$\frac{\partial}{\partial t}\,\mathrm{s_{c,yy}}\,(t,x,y) \quad = \quad -u(x,y)\frac{\partial}{\partial x}\,\mathrm{
[revised manuscript text omitted]

---

## Author Comment (AC2)

First of all, we would like to thank the referee for his/her review on our paper and for giving us the opportunity to improve our paper. We added our acknowledgements to the referee in the new manuscript.

Now, we organized the answer to the comments as follows. First, we list some changes afford to the manuscript then detail our answers to the questions raised by the referee.

**List of changes for the revision**

*Minor changes*

There was an error of sign in the definition of the transport dynamics shown in Fig 9, where the trend should have been defined as minus the velocity times the gradient of concentration. This has been corrected.

*Differences between the two versions of the manuscript*

To facilitate the comparison between the two version of the manuscript, a companion version of the manuscript lists all the modifications where old (new) statements are in red (blue). But the line numbers will refer to the revised version of the manuscript (not to the companion version).

**Answer to the question of the referees**

We copied your commentary in italics below, we reply in normal blue font.

Major comments

*1. "My main comment is that the authors jump directly from section 2.3 into the example of VLATcov models without detailing further the theoretical framework of PKF, making it hard for the reader to conceptualize the approach. For instance, maybe this is a notation issue, but it took me some times to realize that V and s (or g) were the set of parameters p_i of section 2.3.*
*The idea of explaining things with an example is of course a good one, but here I think that simply giving the references at line 123 for the theoretical framework is not enough.*
*Also, the organization of the sections forces the reader to go back and forth between section 2 and 3 to understand what is going on.*
*I suggest to the authors to rewrite sections 2 and 3 in a more streamlined fashion."*

In this version, we make the link between the different parts explicit so to facilitate the reading. This is made at different places as follows:

- We introduced some example of covariance model in the extension of Sec. 2.3, which now end with the introduction of the notation that are used in section 3. So to connect sec. 2.3 and sec. 3.
- We rephrased the end of section 3.1 (l181-184):

"Hence, using the notation introduced in Sec. 2.3, a VLATcov model is a covariance model, $P(P)$, characterized by the set of two parameter fields, $P = (p_1, p_2)$, given by the variance field, and by the anisotropy field – the latter being defined either by the metric-tensor field $g$ or by the aspect-tensor field $s$ – i.e. $P = (V, g)$ or $P = (V, s)$. Said differently, any VLATcov model reads as $P(V, g)$ or $P(V, s)$."

*Intermediate-order comments and questions*

*2. Line 73: "The connection between the Markov process and the parameter dynamics is obtained using the Reynolds averaging technique."Could you provide a citation for the Reynolds averaging technique?*

A reference to the ensemble-average approach has been included (Chap. 4 of *Turbulence in Fluids,* Lesieur).

*3. Sentence line 185 to 187: "In contrast to the matrix dynamics of the KF, the PKF approach is designed for the continuous world, leading to PDEs for the parameter dynamics in place of ODEs Eq. (8) for the full matrix dynamics."*
*What if Eq. (2) is used instead of Eq. (1)? Is there still an advantage to use PKF in this case? More generally, the authors seems to consider also systems like Eq. (2), but then only focus on PDEs. Does that means that SymPKF don't handle such kind of systems? If it does, I would have like an example with a system like Eq. (2). In particular, how to handle SymPKF in this case? Maybe that could be shown in an Appendix.*

While this is not exactly the point you are referring to, the situation where the dynamics takes the form of an ODE can be observed in section 4.6 dedicated to the multivariate situation. Actually, the rhs of Eq.(32) is an ODE. So, the PKF for this part of the dynamics leads to the dynamics of the variance/cross-covariance similar to Eq. (8a). Since $V_{AB}$ equals $V_{BA}$, the PKF provides three equations in place of the four equations of the matrix form. This give you an example of how SymPKF can handle the ODE part for multivariate fields. We mentioned this in l500:
"Note that by tacking into account the multivariate situation with the dynamics of the cross-covariance, the multivariate PKF hybridizes the continuous with the matrix form Eq (8a), which corresponds here to the dynamics of the variances ($V_A$, $V_B$) and the cross-covariance $V_{AB}$"

Now, concerning your point, that is, when the dynamics is given by a true ODE, e.g. a Lorenz 1963 that governs the time evolution of three scalars (which only depend of the time), SymPKF is not designed to handle such dynamics and will crash. SymPKF is only for univariate/multivariate fields over a domain which corresponds to the framework of our research.

*4. line 187: "For the above mentioned scalar fields, introduced is the computation of the algorithmic complexity in section 2.1, the cost of Eq. (16) is O(n)."*
*Please check this sentence. Also I think that the algorithmic complexity is not mentioned in Sec. 2.1 but rather in Sec. 2.2.3*

In the new extension of Sec. 2.3 we detail the cost of the PKF, hence, the appropriate reference is now to the section 2.3 and suppressed the sentences l 187 that was unclear. Thank you.

*5. Section 4.3.2 :*
*Can the closures considered be related to closures found in parameterization scheme?*

We think that the way we close the PKF dynamics is similar to the parametrization while there is no explicit connection with the parametrization we know (for instance in parametrization of the

atmospheric boundary layer). But the nature of the parametrization is different here. In turbulence, the closure is related to the non-linearity, while for the PKF it can be related to linear processes are shown here in the Burgers equation for which the need of a closure is due to the diffusion (this has been detailed in our contribution on the Burgers equation, P18).

*Minor Comments*

*6. line 310: 'Latter' ? Please check this sentence.*

'Latter' has been replaced by 'Thereafter'

*7. line 445: 'lager' -> 'larger'*

This has been corrected

*Recommendation about the code*
*This is not part of the article review but rather a few technical comments about the code*
*to make it better in the long end:*

*8. I advise the authors to develop a proper documentation for their package API. This will*
*encourage and help further collaboration on the code.*

We agree with the referee. Some of our students are now using the code so it is quite stimulating and their feedbacks will be useful to improve this point. We hope this will facilitate the collaboration with the community on this new KF implement.

*9. In the same vein, the code could be more systematically commented.*

This will be done. Thank you.

[revised manuscript text omitted]

---

## Author Response (AR2)

**Answer to the question of the referees**

We copied your commentary in italics below, we reply in normal blue font.

The typos mentioned by the referee have been corrected.
We answer here to the point that needs to be precised.

*1. "Line 136: What are P16 and P18?"*

It is introduced some lines above, nothing to change here.

*2. "Line 147: "but is not enough" => "but not enough" (Also what does "enough" mean precisely here is not very clear to me.)"*

It has been replaced by "but not able to.."

*3. "Lines 148-152: This sentence is too long and should be separated into multiple ones."*

It has been separated in multiple sentences.

*4. "Line 152-153: "Hence ..." I have some trouble following the logic between this sentence and the previous ones. Do you mean that for instance the VLATcov model represent a good and tractable alternative to the previously mentioned models?"*

We rephrased the sentences by merging it with the next sentence as follows:
   "Hence a covariance model adapted for the PKF should be able to represent realistic correlations and be such that the dynamics of the parameters can be computed \eg a covariance model defined by parameters in grid-points. To do so, we now focus on the PKF applied to a particular family of covariance models, whose parameters are defined in grid points by the variance and the anisotropy fields, that is $\Pset=(V,\bg)$ where $\bg$ will denotes the local anisotropy tensor of the local correlation function."

*5. "Line 504: Sorry but I am not sure to understand "hybridizes the continuous". When you say "the continuous", it is related to the continuous nature of the fields?"*

We precise that "the continuous" is related to the continuous nature of the fields:

[revised manuscript text omitted]